

# Improved ELMv1-ECA Simulations of Zero-Curtain Periods and Cold-season CH₄ and CO₂ Emissions at Alaskan Arctic Tundra Sites

Jing Tao[1,2], Qing Zhu[1], William J. Riley[1], Rebecca B. Neumann[2]

[1]Climate and Ecosystem Sciences Division, Lawrence Berkeley National Laboratory, Berkeley, CA, 94720, USA

[2]Department of Civil and Environmental Engineering, University of Washington, Seattle, WA, 98195, USA

*Correspondence to*: Jing Tao (JingTao@lbl.gov)

**Abstract.** Field measurements have shown that cold-season methane ($CH_4$) and carbon dioxide ($CO_2$) emissions contribute a substantial portion to the annual net carbon emissions in permafrost regions. However, most earth system land models do not accurately reproduce cold-season $CH_4$ and $CO_2$ emissions, especially over the shoulder (i.e., thawing and freezing) seasons.

Here we use the Energy Exascale Earth System Model (E3SM) land model version 1 (ELMv1-ECA) to tackle this challenge and fill the knowledge gap of how cold-season $CH_4$ and $CO_2$ emissions contribute to the annual totals at Alaska Arctic tundra sites. Specifically, we improved the ELMv1-ECA soil water phase-change scheme, environmental controls on microbial activity, and cold-season methane transport module. Results demonstrate that both soil temperature and the duration of zero-curtain periods (i.e., the fall period when soil temperatures linger around 0°C) simulated by the updated

ELMv1-ECA were greatly improved, e.g., the Mean Absolute Error in zero-curtain durations at 12 cm depth was reduced by 62% on average. Furthermore, the simulated cold-season emissions at three tundra sites were improved by 84% and 81% on average for $CH_4$ and $CO_2$, respectively. Overall, $CH_4$ and $CO_2$ emitted during the early cold season (Sep. and Oct.), which often includes most of the zero-curtain period in Arctic tundra, accounted for more than 50% of the total emissions throughout the entire cold season (Sep. to May). From 1950 to 2017, both $CO_2$ emissions during the 12 cm depth zero-

curtain period and during the entire cold season showed increasing trends, for example, of 0.26 gC m⁻² year⁻¹ and 0.38 gC m⁻² year⁻¹ at Atqasuk. This study highlights the importance of zero-curtain periods in facilitating $CH_4$ and $CO_2$ emissions from tundra ecosystems.



## 1 Introduction

Cold-season carbon emissions from the Arctic tundra could potentially offset warm-season net carbon uptake under 21[st] century warming climate (Commane et al., 2017; Oechel et al., 2014; Oechel et al., 2000; Koven et al., 2011; Piao et al., 2008; Natali et al., 2019; Belshe et al., 2013; Fahnestock et al., 1998; Jones et al., 1999). Field measurements have indicated large cold-season $CO_2$ losses over Arctic tundra ecosystems (Oechel et al., 2014; Natali et al., 2019). Also, $CH_4$ emitted from September to May were found to contribute more than 50% of the annual total $CH_4$ emissions from Alaska upland tundra

sites (Zona et al., 2016; Taylor et al., 2018). Despite the importance of cold-season carbon emissions and their sensitivity to changing climate, prevailing earth system land models do not accurately reproduce cold-season $CH_4$ and $CO_2$ emissions and their contributions to the annual budgets, largely because of the poorly understood mechanisms of cold-season soil heterotrophic respiration and therefore uncertain numerical representations (Natali et al., 2019; Zona et al., 2016; Wang et al., 2019; Commane et al., 2017). Thus, it remains challenging to assess the response of permafrost carbon dynamics to

Arctic warming and to predict future annual carbon budgets with current Earth System Models (ESMs).

In ESM land models, soil environment influences soil microbial heterotrophic respiration (HR) and decomposition of soil organic carbon (SOC) mainly through applying prescribed temperature and moisture functions to modify base decomposition rates. These functions, however, rely heavily on empirical or semi-empirical relationships which are highly uncertain (Sierra

et al., 2017; Sierra et al., 2015; Yan et al., 2018; Moyano et al., 2013; Tang and Riley, 2019; Rafique et al., 2016; Bhanja and Wang, 2020; Kim et al., 2019). Specifically, the temperature sensitivities of soil carbon decomposition is often represented with a $Q_{10}$ value (i.e., the increase in respiration rate from a 10°C increase in temperature) that is fixed at 1.5 or 2.0 (Meyer et al., 2018). However, the values of $Q_{10}$ are controversial (Davidson and Janssens, 2006). Some studies found a uniform $Q_{10}$ across biomes and climate zones, e.g., as 1.4 (Mahecha et al., 2010). Other studies demonstrated that $Q_{10}$ varies with

environmental conditions, ecosystem types, and soil texture (Meyer et al., 2018; Graf et al., 2011; Kim et al., 2019), showing a large spatial heterogeneity with generally higher values in the high-latitudinal regions (Zhou et al., 2009). In addition, Wilkman et al. (2018) reported a temporal heterogeneity in $Q_{10}$ over the Alaskan Arctic Tundra and suggested a higher value (e.g., 2.45) for early summer (e.g., June) but lower value (e.g., 1.58 to 1.67) for the peak growing season (e.g., July). Dynamic decomposition temperature sensitivities are also consistent with theory of microbial dynamics (Tang and Riley,

2015). Also, the response of HR to changes in soil moisture is commonly expressed by empirical relationships in ESMs, which vary substantially (Sierra et al., 2015; Yan et al., 2018; Moyano et al., 2013). Although in-situ measurements reveal that microbial respiration occurs under very cold conditions (e.g., even when soil temperature is lower than -15 °C) (Natali et al., 2019; Zona et al., 2016), most process-based models completely shut down microbial activity due to limited liquid water in freezing and subfreezing soils, and few modelling studies have closely investigated the HR-moisture relationships in

frozen conditions.





The strong dependency of $CO_2$ and $CH_4$ emission on soil temperature and moisture in ESM land models (Riley et al., 2011; Koven et al., 2017; Lawrence et al., 2015) requires accurate estimates of these two closely related soil variables, especially in cold regions where both increases and decreases in soil temperature could lead to soil "drying" due to drainage or freezing

processes. However, current land models tend to significantly underestimate soil temperature during the cold season over permafrost regions (Dankers et al., 2011; Tao et al., 2017; Nicolsky et al., 2007; Yang et al., 2018b). One possible reason is that many land models fail to appropriately account for the latent heat released during soil water freezing (Yang et al., 2018a; Nicolsky et al., 2007). Latent heat released during freezing might be sufficient to offset heat conduction towards the surface, thus maintaining the subsurface soil temperature around the freezing point (i.e., 0°C) for weeks or even months during the

fall (i.e., the so-called Zero-Curtain Period; ZCP) (Outcalt et al., 1990). The ZCP conditions allow for continued soil heterotrophic respiration at notable rates, and thus $CO_2$ and $CH_4$ production and emissions from subsurface soils (Kittler et al., 2017; Arndt et al., 2019; Commane et al., 2017). For instance, Zona et al. (2016) reported that a substantial portion of cold season $CH_4$ emissions occurred during the ZCP from Alaskan upland tundra sites. Nevertheless, many land models cannot accurately capture the ZCP length due to their underestimation of soil temperatures, thus underestimating cold-season

emissions of $CO_2$ (Commane et al., 2017) and $CH_4$ (Zona et al., 2016).

We hypothesize that the underestimation of modelled cold-season $CO_2$ and $CH_4$ emissions in ESMs land models primarily results from underestimated soil temperatures during the cold season, the poor representations of environmental controls on heterotrophic respiration in subfreezing soils, and the lack of appropriate representation of cold-season methane transport

processes. Here we apply the Energy Exascale Earth System Model (E3SM) land model version 1 (ELMv1-ECA) (Golaz et al., 2019; Zhu et al., 2019) to explore these hypotheses. We apply ELMv1-ECA to (i) improve simulations of subsurface soil temperatures, ZCPs, and $CO_2$ and $CH_4$ emissions over the permafrost tundra ecosystem; (ii) investigate the underlying processes that influence cold-season carbon emissions from freezing and subfreezing soils, including source characterization and transport pathways; and (iii) estimate historical trends (from 1950 to present) of cold-season $CO_2$ and $CH_4$ emissions at

multiple Alaskan tundra sites.

The paper is organized as follows: (1) We describe the study sites and the data used in the study. (2) We present the theoretical background of essential modules of ELMv1-ECA relevant to this study and our modifications to the model's representations of phase-change, SOC decomposition, and methane dynamics. (3) We then describe the model configuration

and experimental design. (4) We assess the modified phase-change scheme by comparing simulated soil temperatures and ZCPs against observations. (5) With the revised phase-change scheme and methane module, we analyze how the decomposition schemes impact simulated $CO_2$ and $CH_4$ emissions at the site scale. (6) Finally, we summarize the main findings and discuss needed observations and model development to further improve predictability.



## 2 Study Sites and Data

We assembled daily observations of $CO_2$ and $CH_4$ fluxes from 2013 to 2017 at five eddy-covariance flux tower sites in Alaska's North Slope tundra (Figure 1) from the Arctic-Boreal Vulnerability Experiment (ABoVE) project (2015 - 2017) (Oechel and Kalhori, 2018) and Carbon in Arctic Reservoirs Vulnerability Experiment (CARVE) flight campaign (2013 - 2014) (Zona et al., 2016). CARVE $CO_2$ measurements were not available; therefore, monthly winter-time $CO_2$ flux data at the ABoVE towers assembled by Natali et al. (2019) are included to complement $CO_2$ observations from 2013 to 2014. The

five sites include three eddy covariance (EC) towers at Barrow (i.e., the Barrow Environmental Observatory (BEO) tower, the Biocomplexity Experiment South (BES) tower, and the Climate Monitoring and Diagnostics Laboratory (CMDL) tower), one tower at Atqasuk (ATQ) and another at Ivotuk (IVO) which is located at the foothills of the Brooks Range. BES and CMDL are collocated with each other with sensors installed at different heights (i.e., 2 m for BES and 5 m for CMDL). Vegetation at Barrow is mainly moist acidic tundra. Instrument height at ATQ and IVO is 2 m and 4 m, respectively. ATQ is

a well-drained upland site, and the vegetation consists of moist-wet coastal sedge tundra and moist-tussock tundra surfaces. Vegetation at IVO is polar tundra. Table S1 provides basic information including geolocations, vegetation mosaic, and climatologic air temperature at the sites. (Tables numbered with a prefix "S" are include in the supplementary file, which will not be repeated in the following context throughout the manuscript.)

ABoVE and CARVE provide soil temperature and moisture measurements at various depths from 5 cm to 40 cm. The Permafrost Laboratory, Geophysical Institute of University of Alaska Fairbanks (GIPL-UAF), provides daily subsurface soil temperature observations down to various depths at permafrost sites across Alaska(http://permafrost.gi.alaska.edu/sites_map) (Romanovsky et al., 2009). We used the GIPL-UAF permafrost sites that are collocated with the ABoVE sites to complement the ABoVE observations at deeper depths, including BR2 (down to 15 m) and IV4 (down to 1 m). We first

filled missing gaps vertically by fitting a polynomial to the soil temperature profile (Kurylyk and Hayashi, 2016) on a daily scale, then screened out outliers by examining the daily time series. Further, we aggregated both the ABoVE and the GIPL-UAF soil temperature measurements to ELMv1-ECA soil layer node depths using the inverse distance weighting method (Tao et al., 2017), and then averaged the two sets of aggregated observations. We used the assembled subsurface temperature observation datasets to evaluate the ELMv1-ECA simulated soil temperature profiles and the zero-curtain periods.


Due to the discontinuity of observed soil moisture over time and along with the vertical depth, evaluating ELMv1-ECA simulated liquid water content at layer node-depth was limited. We matched soil-moisture observations to the vertically closest model layer, and then evaluated the simulated volumetric fraction of soil liquid water content at layers for time periods during which observations were available. In addition, we used ABoVE soil moisture measurements to derive site-

scale soil porosity and organic carbon content (see Section 3.2).





## 3 Methodology

### 3.1 Modifications to E3SM Land Model (ELM)

The E3SM land model version 1 (ELMv1-ECA) couples essential biogeophysical and biogeochemical processes that solve terrestrial ecosystem energy, water, carbon, and nutrient dynamics (Golaz et al., 2019; Zhu et al., 2019). Figure 2 illustrates
the coupling and interactions among the three components. In the appendix, we describe in detail its subsurface soil thermodynamics, the carbon decomposition module, and the methane module that are of particular relevance to our study. Here we identify the potential problems of ELMv1-ECA that are responsible for the underestimation of cold-season $CH_4$ and $CO_2$ emissions and summarize the modifications made to ELMv1-ECA, emphasizing the model enhancements, shown by the ellipses with red boundaries in Figure 2.

### 3.1.1 Phase Change Scheme

We first improved ELMv1-ECA's numerical representation of coupled water and heat transport with freeze-thaw processes via improving the phase-change scheme. The freeze-thaw processes of soil water within ELMv1-ECA is simulated in a decoupled way, i.e., it solves soil temperatures ignoring the latent heat associated with phase change, determines the mass change of soil water required to adjust the initially solved soil temperature to the freezing point (i.e., 0°C; $T_f$), adjusts the soil
liquid and ice content by mass and energy conservation, and then readjusts temperatures after accounting for the heat deduction or compensation resulted from melting or freezing (see the detailed description in the Appendix A). The underlying assumption here is, taking the freezing process as an example, the available liquid water at the initially solved temperature ($T_i^{n+1}$) will be completely frozen, releasing latent heat ($H_i$) to bring up $T_i^{n+1}$ back to $T_f$. Then, the estimated phase-change rate will be tuned down and the current temperature (i.e., $T_f$) will be readjusted if the to-be-increased ice mass
is larger than the required mass change ($-H_m$) (see (Eq. A4) in the Appendix A), which, however, only occasionally occurs. When the liquid water available to be frozen becomes small enough, the released latent heat is not sufficient to compensate for the required energy deficit ($T_f - T_i^{n+1}$), and then the freezing process stops. Consequently, the model freezes soil water quickly, resulting in an underestimated duration of the soil water phase-change processes and the zero-curtain periods, and also cold-biased winter temperatures (Nicolsky et al., 2007; Yang et al., 2018a).


Here, we employed a phase-change efficiency and the temperature of the freezing-point depression to effectively solve the problem of overestimating phase-change rates within the current ELMv1-ECA modelling structure. These modification factors are explained below. The phase-change efficiency, introduced by Le Moigne et al. (2012) and adopted by Masson et al. (2013) and Yang et al. (2018a), introduces the dependency of available liquid water on the phase-change rate (Le Moigne
et al., 2012). The phase-change efficiency for freezing, $\varepsilon_{liq,i}^n$ (see (Eq. A7)), is identical to the degree of moisture saturation, or the volumetirc fraction of soil liquid water content (i.e., $Sf_{liq,i}^n = \theta_{liq,i}^n / \theta_{sat,i}^n$ where $\theta_{liq,i}^n$ is soil liquid water content and





$\theta_{sat,i}$ is porosity). The underlying assumption here is that the liquid water of soil resists freezing as the freezing process proceeds and $Sf_{liq,i}^n$ decreases, analogous to how dry soils resist getting drier due to capillary force. We applied the phase-change efficiency to the initially estimated energy and mass change involved, i.e., $H_i$ and $H_m$ (see (Eq. A4) in the Appendix) when freezing or thawing process occur.

As in Nicolsky et al. (2007) and Yang et al. (2018a), the occurrence of a phase-change process is then determined by the temperature of the freezing point depression (i.e., an virtual temperature, see (Eq. A8)) instead of $T_f$. The virtual freezing point depression temperature is reversely derived from the freezing point temperature-depression equation (Fuchs et al., 1978; Cary and Mayland, 1972). With an upper limit as $T_f$, the virtual temperature describes the lowest temperature that can hold current liquid water content in the freezing soils. That is, the soil temperature has to be lower than the current virtual temperature to allow the freezing process to occur further.

We describe in detail the revised phase-change scheme in the Appendix A. In short, we improved the phase-change scheme of ELMv1-ECA by incorporating two modifications: 1) applying a phase-change efficiency to implicitly account for the heat compensation/deduction to the system from latent heat released/absorbed by soil water freezing/melting, and 2) replacing the constant freezing point with the temperature of the freezing point depression, as a virtual temperature, to determine the occurrence of phase change in subfreezing soils.

### 3.1.2 Environmental Modifiers to the Decomposition Rate

We revisited ELMv1-ECA's representation for soil heterotrophic respiration dynamics in subfreezing soils and then scrutinized the environmental scalars of soil temperature and moisture. Within ELMv1-ECA's decomposition cascade model, the environmental factors that impact the decomposition rates of soil organic matter include soil temperature ($f_T$), soil moisture ($f_W$), oxygen stress ($f_O$) and a depth scalar ($f_D$) (See Appendix B). Within freezing and subfreezing soils, the soil water potential is related to temperature through the freezing point depression equation (Niu and Yang, 2006). The current moisture factor $f_W$, therefore, can predict zero respiration rates for subfreezing soils given a minimum soil water potential $\psi_{min}$, as shown by Figure S1a in the supplementary file. (Figures numbered with a prefix "S" are include in the supplementary file, which will not be repeated in the following context throughout the manuscript.) We thus imposed a minimum threshold ($f_{W\_min}$) to prevent zero respiration within the active layer when soil becomes subfreezing during cold-season months (Figure S1b).

For wet soils, the factor that primarily limits the decomposition rates is oxygen availability (Sierra et al., 2017), since increases in soil moisture result in decreased dissolved oxygen. ELMv1-ECA approximates oxygen stress ($f_O$) as a ratio of available oxygen to the demand by decomposers, which, however, is highly uncertain and unstable (Oleson et al., 2013).





Adapting the concept and formulation of Yan et al. (2018), we incorporated oxygen stress into the moisture scalar to account for the inhibition of decomposition in wet anoxic conditions. The revised form of the moisture scalar $f_W^*$ (Eq. B11) gradually decreases when the degree of saturation exceeds an optimal wetness threshold ($Sf_{op}$) that represents the most favorable soil moisture condition for decomposition, as shown by Figure S1b. We also tested several modified moisture schemes with different shape parameters ($b$ in Eq. B11) and optimal wetness thresholds and minimum thresholds ($Sf_{op}$ and $f_{W\_min}$ in Eq. B11). When using the modified moisture scalar with the built-in oxygen stress, the total environmental impacts on decomposition, i.e., $f_{total} = f_T f_W f_O f_D$ will be modified accordingly as $f_{total} = f_T f_W f_D$ to avoid double-counting of the oxygen stress.

ELMv1-ECA uses a $Q_{10}$-based standard exponential function to account for the temperature effect on SOC decomposition (Eq. B9), with $Q_{10}$ as 1.5 and $T_{ref}$ as 25°C. Here, rather than striving for a single value of $Q_{10}$, or a spatial map of $Q_{10}$ as discussed in the introduction, we seek an optimized scheme at the site scale and a generic scheme at the regional scale for the total environmental modifier ($f_{total}$) that combines moisture ($f_W$) and temperature ($f_T$) sensitivity. Specifically, we assembled and tested 200 cases of $f_{total}$ using the newly modified moisture scalars with different parameters $b$, $Sf_{op}$, and $f_{W\_min}$, temperature scalars with different values of $Q_{10}$ and $T_{ref}$, and a variety of other empirical moisture and temperature functions, as documented by Sierra et al. (2015). A full list of the specific moisture and temperature scalars used is provided in Table S2.

### 3.1.3 Cold-season Methane Process

The ELMv1-ECA methane model solves the reaction and diffusion equation for $CH_4$ and $O_2$ fluxes with the Crank-Nicholson method. It includes the representations of $CH_4$ production, oxidation, and three pathways of transport, including aerenchyma tissues, ebullition, aqueous and gaseous diffusion (Riley et al. (2011)). A short description of the ELMV1-ECA methane module is provided in Appendix C. The ELMv1-ECA methane model has been found to underestimate cold-season methane emissions over northern wetlands (Xu et al., 2016). The modifications to the phase-change scheme impact simulations of soil water and heat transfer (3.1.1); the changes in environmental scaler affect substrate availability (3.1.2). Both (3.1.1) and (3.1.2) influence carbon decomposition and soil heterotrophic respiration (Figure 2), and could potentially lead to improvements in simulated $CO_2$ and $CH_4$ production, but not necessarily $CH_4$ emissions which are also controlled by transport mechanisms (Figure 2). Thus, we further refined the cold-season methane transport processes.

Here, we first modified the ELMV1-ECA $CH_4$ transport mechanism in cold seasons by mimicking possible pathways for $CH_4$ emissions from freezing and subfreezing soils. Specifically, we mimic the emissions from ice cracks by plant aerenchyma transport (Zona et al., 2016), approximating the gas diffusion through ice cracks to the similar mechanism of diffusion through the aerenchyma tissues. Although in-situ experiments demonstrated that during winter, produced $CH_4$ in



frozen soils is predominately emitted to the atmosphere through vascular plants aerenchyma tissues (e.g., Kim et al., 2007), here we integrate the possible transport pathways including ice cracks and remnants of aerenchyma tissues together through equation (Eq. C14).

During the cold season over the tundra ecosystem, snow on the land surface provides strong resistance to $CH_4$ transport to the atmosphere in ELMv1-ECA, as shown in Figure 2. But in reality, studies have shown methane can diffuse through snowpack at varying rates (Kim et al., 2007). We thus decreased snow resistance at the upper boundary by introducing a new scale factor when snow is present. Also, in ELMv1-ECA, the aqueous diffusion coefficients in freezing and subfreezing soils below the water table are based on the volumetric fraction of the liquid water content, which is quite small (i.e., the

supercooled liquid water) and thus limits diffusion. We revised the formulation (Eq. C15), assuming a higher scaling factor for frozen soils ($f_{frzsoil}$) upon sensitivity experiments (not shown). Table 1 summarized all the specific modifications made to ELMv1-ECA. These modifications involve new parameters that are all tuneable and can be systematically optimized via calibration. Here, we seek to reproduce the first-order cold-season process relevant to this study with these default formation and values listed in Table 1.

**3.2 Climate Forcing, Model Configuration, and Experiment Design**

We conducted transient simulations at 30-minute temporal resolution driven by climate forcing from 0.5°×0.5° CRU JRA (Harris, 2019) from 1901 to 2017 at the four Alaska tundra site locations. Before the transient simulation, we conducted a 200-year Accelerated Decomposition (AD) spin-up period followed by a 200-year regular spin-up period (Koven et al., 2013b; Zhu et al., 2019) to initialize land carbon pools. Spin-up simulations start from a wet and cold condition. Specifically,

sub-surface temperatures were initialized as 274 K for the 1[st] to 5[th] soil layers, 273 K for the 6[th] to 10[th] layer, and 272 K for the 11[th] to the 15[th] layer, and volumetric soil water content was initialized fully saturated for all layers. In this manner, consistent vertical soil water content profiles were built in over the permafrost regions.

Baseline simulations were conducted with ELMv1-ECA default physics, parameters, and surface datasets, i.e.,

OriPC_OriDecom_OriCH4 using original phase-change scheme, original decomposition scheme and methane module (Table 2). To improve the model representation of the site-level soil environment, we first examined the global soil organic matter data at the ABoVE sites by evaluating ELMv1-ECA simulated subsurface soil temperature with the topsoil temperature prescribed to observations (as did in Tao et al., 2017). Using the top soil layer as the upper boundary, the modelling system excluded potential errors induced by inaccurate meteorological forcing and vegetation cover that impact the simulation of

heat transfer from the atmosphere to the shallow soil (Tao et al., 2017). Then, the accuracy of simulated soil subsurface temperature is directly determined by the factors impacting heat transfer along the "shallow-to-deep soil" gradient (Koven et al., 2013a), e.g., soil thermal properties which are mostly determined by SOC content (Tao et al., 2017; Lawrence and Slater, 2008). Results well reproduced the subsurface soil temperatures except at IVO, where summer soil temperatures were



notably overestimated (see Figure S2a). This result indicates that the SOC content at IVO was too small, leading to a large
thermal conductivity, small soil porosity, and small heat capacity, altogether resulting in fast penetration of heat into the
subsurface soil during summer (Tao et al., 2017; Lawrence and Slater, 2008). Thus, we derived the organic matter density at
IVO based on ABoVE soil moisture data through a linear relationship between SOC content and soil porosity (i.e., Equation
3 in Lawrence and Slater (2008)), assuming the observed maximum volumetric water content was porosity (see Figure S3
for details). With the newly derived profile of soil organic matter density at IVO, the simulation showed large improvements
in summer soil temperatures compared to that using the original global carbon dataset (see Figure S2b). The derived SOC
content is also consistent with the soil survey data reported in Davidson and Zona (2018). Hereafter, the simulations at IVO
presented in this paper use the newly derived organic carbon data without repeated clarification.

The representative spatial scale of the eddy flux tower is small compared to the grid cell of global surface datasets and the
climate forcing data used by ELMv1-ECA, although the forcing dataset was interpolated to the site scale with a bilinear or
nearest-neighbor method. The site-scale vegetation cover also shows a large diversity of vegetation types according to the
detailed vegetation survey at ABoVE flux tower footprints obtained in 2014 (Davidson and Zona, 2018). We analyzed the
vegetation composition from the closet survey plot to the flux tower and examined the rationality of ELMv1-ECA's
percentage of plant type function (PFT) for the site-scale simulation. We confirmed that ELMv1-ECA's PFT dataset was a
good compromise between representing the site-scale ecosystem and other global parameters and surface datasets within
ELM. The simulated saturated and unsaturated $CH_4$ emissions were weighted with the estimated inundation fractions at the
footprint of ABoVE eddy-covariance flux towers (see details in (Xu et al., 2016)) in order to compare simulated $CH_4$
emissions with ABoVE measurements at the site scale.

Table 2 lists the experiments conducted in this study. We modified each model component (i.e., the heat transfer model,
carbon decomposition model, and methane model) serially. For the temperature- and moisture-dependency functions, we
analyzed 200 environmental modifiers within the carbon decomposition module and identified an optimal scheme for each
site and a generic scheme that can be applied for the regional simulation over Alaskan North Slope tundra (see next section).

**3.3 Evaluation Method and Trend Analysis**

We define the early cold season as September and October, the cold-season period as September to May which includes the
two shoulder seasons (both thawing and freezing) as consistent with Zona et al. (2016), and the warm season from June to
August. We define the zero-curtain period (ZCP) as the set of successive days when the soil temperature is within the range
of [-0.75°C, 0.75°C] starting in fall (i.e., the freezing season) based on Zona et al. (2016). We computed the ZCP duration
for each soil layer every year from 1950 to 2017 and estimated the historical trend as the regression slope between ZCP
duration and time. Similarly, we estimated the trends of cold-season $CH_4$ and $CO_2$ emissions through linear regression



analysis. A p-value of 0.05 is used to determine if the computed trend is statistically significant. Results vary with soil depths; thus, we choose a common modelling depth, i.e., 12 cm, which locates within the active layer for all the sites, to give an example.


We used Nash-Sutcliffe Efficiency (NSE) (Nash and Sutcliffe, 1970) to examine the performance of the ELMv1-ECA simulated time series of $CH_4$ and $CO_2$ net fluxes in comparison with assembled observations (Section 2) at the monthly time scale. The NSE ranges from negative infinity to one, calculated as Eq. (1):

$$NSE = 1 - \left(\frac{1}{N}\sum_{t=1}^{N}\left(\hat{E}_t - O_t\right)^2\right)/\sigma_o^2, \tag{1}$$

where t means monthly time step, N is the total number of time steps, $\hat{E}_t$ and $O_t$ is simulated and observed flux at time step t,
respectively; and $\sigma_o$ is the standard deviation of observations. Note we only used observed monthly averages when the number of daily observations was more than 20 days. The model performance is generally considered satisfactory with an NSE > 0.50 (Moriasi et al., 2007), and perfect with an NSE as one. To simultaneously evaluate $CH_4$ and $CO_2$ fluxes, we combined both $NSE_{CH4}$ and $NSE_{CO2}$ in the form of $dist = \sqrt{(1 - NSE_{CH4})^2 + (1 - NSE_{CO2})^2}$, representing the distance from $(NSE_{CH4}, NSE_{CO2})$ to (1, 1) in a coordinate plane with x-axis as $NSE_{CH4}$ and y-axis as $NSE_{CO2}$. The optimal simulation
thereby is the one having the shortest distance to the ideal scenario (1, 1). We also define a satisfactory model performance in terms of simulating $CH_4$ and $CO_2$ fluxes as the case with both $NSE_{CH4}$ and $NSE_{CO2}$ larger than 0.5. The generic scheme then is the common satisfactory scheme that provides the best overall performance for all the sites.

To evaluate ELMv1-ECA simulated soil temperature and moisture, we calculated the RMSE for each soil layer,
i.e., $\sqrt{\sum_{t=1}^{N}\left(\hat{E}_t - O_i\right)^2/N}$ where the $\hat{E}_t$ and $O_t$ is simulated and observed soil temperature or moisture, respectively, and t is a daily time step. We used the Mean Absolute Error (MAE, $i.e., \frac{1}{N}\sum_{t=1}^{N}\left|\hat{E}_t - O_i\right|$ to assess the simulated duration of ZCP of each soil layer. Note that, depending on the amount of soil liquid water content, the whole course of the freezing process may or may not entirely fall into the ZCP, i.e., the ending time of ZCP does not necessarily align with the end of the freezing process. The onset of freezing, though, is always later than the starting day of the ZCP, and the main course of the freezing
process is still within the ZCP.

Here the modelled active layer thickness (ALT), i.e., maximum thaw depth during an annual cycle, is computed as the bottom depth of the deepest thawed soil layer (i.e., with a maximum annual temperature above 0°C) further extended down to the possible non-frozen fraction of the layer below, as in Tao et al. (2019; 2017). We only derived the length of ZCP for
soil layers with a maximum annual temperature above 0°C since limited phase-change processes occur in deeper layers. Then, the soil layers containing or below the permafrost table have a zero-day ZCP. We computed the MAE of ALT





simulated with both original (OriPC) and the new phase-change (NewPC) scheme. Also, we computed the relative improvement in simulated soil temperature (Ts) and ZCP compared to the baseline results. Specifically, we calculated 100% × (RMSE_Ts_OriPC – RMSE_Ts_NewPC) / RMSE_Ts_OriPC and 100% × (MAE_ZCP_OriPC – MAE_ZCP_NewPC) /

MAE_ZCP_OriPC to quantify the enhancement by employing the new phase-change scheme.

In general, we use NSE to evaluate the model's performance in capturing seasonality (i.e., time series of $CH_4$ and $CO_2$ net fluxes) and use RMSE and MAE to assess the model's capability in simulating the magnitudes of soil temperature, moisture saturation, and ZCP durations.

**4 Results and Discussion**

**4.1 Evaluation of Soil Temperature and Zero-curtain Period**

We first evaluated the simulated daily soil temperature profiles against the observations from ABoVE and GIPL-UAF at the four site locations. Then, we examined improvements in simulations of soil temperature, soil moisture, and the durations of ZCPs by employing the newly revised phase-change scheme (i.e., "NewPC_OriDecom_OriCH4"; Table 2).


Results for the BES/CMDL and IVO site are shown in Figure 3; results for other sites are shown in supplementary Figure S4. At BES/CMDL, the baseline (i.e., "OriPC_OriDecom_OriCH4"; Table 2) simulated soil temperatures (Ts) with the default phase-change scheme (Ts_OriPC; cyan lines; Figure 3a) decrease rapidly in fall due to the overestimated freezing rate (i.e., the slope of decreasing liquid water fraction), notably underestimating the duration of the ZCP (greenish shaded

area). Consequently, liquid water saturation (S_f_OriPC, green lines; Figure 3a) quickly drops to a lower bound (i.e., the supercooled liquid water content divided by porosity), and the freezing process generally completes within a short period (days for top layers to one month at the most for deeper layers). The baseline model soil temperature drops (Ts_OriPC) sharply after the freezing process ends (i.e., S_f_OriPC decreases to the lower bound). In contrast, the new phase-change scheme effectively slows freezing rates, showing relatively smaller slopes of decreasing liquid water saturation (S_f_NewPC;

magenta lines; Figure 3a) within the ZCPs than the baseline simulation (S_f_OriPC; green lines) especially in the 4th and 5th layer. Hence, the gradually released latent heat maintains soil temperatures around the freezing point for a longer period (Ts_NewPC; blue lines; Figure 3a), effectively extending the ZCPs (blue shaded area) which agree better with observations than the baseline results. The ZCP duration increases with depth and can extend into December for deep soil layers. Similarly improved performance was found at the BEO and ATQ sites (supplementary Figure S4). At IVO, however, while

the new phase-change scheme greatly improved simulated results relative to the baseline simulation (Figure 3b), the model still slightly underestimated ZCP durations and also underestimated winter (December to April) soil temperature (blue vs. red). This result at IVO is consistent with the underestimation of late-season soil liquid water available to be frozen, and thereby to release sufficient latent heat (Figure S5).





Simulated ZCP durations with the revised phase-change scheme (NewPC) demonstrated notable improvements over the baseline (original) phase-change scheme (OriPC) (solid circles vs. open diamonds) (Figure 3), showing greatly reduced mean absolute errors (MAEs) (Table 3). For example, at 12 cm depth ($4^{th}$ layer), the relative improvements in MAE of the ZCP durations were 65%, 65%, 66%, and 50% for the four site locations (Table 3). The largest improvement in MAE was as large as 65 days for the $6^{th}$ layer at BES/CMDL, with a relative improvement of 84% (Table 3). This large improvement

stems from the better-estimated ALT at this site; the OriPC simulated $6^{th}$ layer temperature remained below freezing, leading to a zero-day ZCP (diamonds on the x-axis in Figure 4). The new phase-change scheme not only improved simulation of the ZCP and cold-season soil temperatures, but also affected the warm season dynamics and thus ALT estimates. As Figure 4 indicates, the NewPC improved simulated ALTs at all four site locations with reduced bias in multi-year averaged ALT, resulting in more reasonable ZCP durations for the $6^{th}$ layer (and also $7^{th}$ layer for IVO), while the baseline results were zero

days.

The deeper active layer simulated by NewPC implies more soil water storage capacity, resulting in lower soil moisture in shallow soil layers and higher soil water in deep layers ($S_f\_NewPC$; magenta lines; Figure 3) compared to baseline results. The changes in soil liquid water content, in turn, impact phase-change and soil temperature simulations. Comparison with

the observed soil liquid water content reveals a better agreement with observations (Table S3). For example, at ATQ (Figure S6), the RMSEs of the liquid water content were reduced by 5.4%, 35.3%, 42.6%, and 25.4% for the $3^{rd}$ through $6^{th}$ layers, respectively (Table S3).

The changes to model representations of phase change led to large reductions in soil temperature bias. The relative

improvements in RMSE of simulated soil temperatures during Sep. and Oct. (i.e., the two months that the ZCPs usually cover), generally increased with depth for surface layers (within about 20 cm of the surface, i.e., $1^{st}$ to $4^{th}$ layer), and were above 80% for the intermediate layers ($5^{th}$ to $8^{th}$) at all the sites (Figure 5). At the two Barrow sites where observed soil temperatures were available, the relative improvements for the deepest ($13^{th}$) layer were 72.6% and 71.1%, on average, for the early winter and annual cycle, respectively. Therefore, incorporating the new phase-change scheme also resulted in

improved bottom temperature boundary conditions, which is critical for accurately simulating permafrost dynamics (Sapriza-Azuri et al., 2018). Improvements between Septemper and December and the whole annual cycle also increased with soil depth, showing site-averaged reductions in RMSEs ranging from 47% to 63% and from 36% to 46% for the two periods, respectively. The whole cold-season period (Sep. to May) showed, on average, 44% to 53% reduction in RMSEs from the $1^{st}$ to $6^{th}$ layer at relatively warmer sites (i.e., ATQ and IVO), and from 19% to 69% for the top 13 layers for the two Barrow

sites.





Soil temperatures were still slightly underestimated during the thawing season (i.e., May) at all four sites, showing later onset of thawing indicated by the timing when warming soil temperatures cross 0°C and soil moisture starts to rise (Figure 3). One possible reason for this bias is the lack of representation of advective heat transport. That is, the model does not

represent the heat of spring rain that is advectively transported into soils (Neumann et al., 2019; Mekonnen et al., 2020); nor does it account for advective heat transport associated with water fluxes in subsurface soils after the spring-rainwater mix with existing cold liquid water in soils. Also, after the freezing process ends, simulated deeper soil layer temperatures were underestimated (e.g., December through April). This bias might be caused by underestimated snow depth (not shown) resulting from inaccurate forcing (particularly snowfall), land cover, microtopography, and/or wind-blown snow

redistribution.

The improved simulations of soil temperature, liquid water content, and ZCP duration greatly impacted soil HR and methane production (Figure 6). Increases in the baseline modeled HR and $CH_4$ production resulted from changes in soil temperature and moisture (Figure 6b1 and b2 vs. Figure 6c1 and c2) and mainly occured within the two-dimensional "zero-curtain zone"

across the vertical soil profile spanning multiple months, i.e., from September to November (Figure 6c1). However, still very small HR and $CO_2$ and $CH_4$ production were predicted during the following cold season months (Figure 6c3 and c4) due to the moisture scalar for subfreezing soils estimated by ELMv1-ECA's original moisture-dependency function on decomposition (Eq. B10), as discussed in Section 3.1.2. In addition, the sharp decreases of HR and $CH_4$ production around the end of September were caused by the dramatically increased oxygen stress (i.e., the decreased oxygen scalar) to

decomposition when freezing began (Figure 6c3 and c4). By replacing the original moisture scalar with the modified soil moisture-dependency function scheme-2 with oxygen stress ((Eq. B11), also see Figure S1) along with the modified total environmental modifier, both the near-zero respiration and sharp drawdown trends in HR and $CO_2$ and $CH_4$ production were greatly alleviated (Figure 6c3 and c4 vs. Figure 6d3 and d4). In the next section, we analyze 200 environmental modifier schemes to the base decomposition rate (Table S2) that assembled commonly used empirical soil temperature- and moisture-

dependency functions as documented by Sierra et al. (2015) and the modified functions proposed in this study.

### 4.2 Evaluation of CO₂ and CH₄ Emissions

Here we evaluate the simulated monthly $CO_2$ and $CH_4$ fluxes at the site scale against EC tower observations. Figure 7 displays the NSEs of 200 ELMV1-ECA ensemble simulations with different combination of temperature and moisture

scalers on soil decomposition, i.e., "NewPC_NewDecom_NewCH4" (grey dots) (see configurations in Table 2). (Daily time series of all the simulations are provided in Figure S7). The failure of simulated $CH_4$ emissions to capture the methane seasonality at IVO (as indicated by Figure S7) might occur because of the lack of 1) a reasonable wetland module that can adequately account for inundated hydro-ecological dynamics, 2) advective heat transport at the air-ground interface through rainfall infiltration and within subsurface soils through water transfer, and 3) the geological micro-seepage emission of $CH_4$,



as reported in previous studies (Anthony et al., 2012; Etiope and Klusman, 2010; Russell et al., 2020). For instance, Lyman et al. (2020) showed large temporal variability of $CH_4$ at natural gas well pad soils, similar to the observations at IVO (Anthony et al., 2012). Controlled experiments (not shown) that imposed observed soil temperature and moisture into the modelling system at all the layers with observations available do not demonstrate improvement for the simulation of $CH_4$, although showing better performance for $CO_2$. These results confirm that impacts from the soil environment (e.g., soil

temperature and moisture) within the current water and heat transfer framework cannot explain the seasonal variability of $CH_4$ emissions. Thus, the three mechanisms discussed above (i.e., wetland dynamics, advective heat transport, and geological micro-seepage $CH_4$ emission) currently missing in our model are likely necessary to simulate $CH_4$ emissions at this site and we therefore do not include analysis at IVO in the following sections.

The improved phase-change scheme, and thus improved simulations of ZCP durations and soil temperature and moisture, resulted in greatly improved performance for $CO_2$ emissions at BES/CMDL and BEO, and slightly better performance for $CH_4$ emissions at ATQ, compared to the baseline (cyan for "NewPC_OriDecom_OriCH4" vs. green for baseline; Figure 7), even though the carbon decomposition and methane modules remained the same. Incorporating the revised $CH_4$ model (discussed in section 3.1.3) improved simulated $CH_4$ emissions at BES/CMDL, BEO, and ATQ (blue for

"NewPC_OriDecom_NewCH4" vs. cyan for "NewPC_OriDecom_OriCH4"), especially during the cold season (Figure 8). The improved NSEs for $CH_4$ emissions mainly resulted from increased emissions over early winter (Sep. and Oct.) and slight but persistent enhancements throughout the rest of the cold season (blue in Figure 8), which were related to our modifications to $CH_4$ transport mechanisms. Further, with the identified optimal scheme of environmental modifiers to the base decomposition rate, results demonstrate substantial improvements to the simulation of cold-season $CO_2$ and $CH_4$

emissions compared to baseline results (yellow vs. others; i.e., shortest distance from ($NSE_{CH4}$, $NSE_{CO2}$) to (1, 1)). Among the common schemes providing good performance for both $CO_2$ and $CH_4$ emissions (i.e., both $NSE_{CH4}$ and $NSE_{CO2}$ larger than 0.5, indicated by the gray dots within the boxes in Figure 7), we identified a generic scheme of environmental modifier to the decomposition rate by selecting the common scheme that provided best overall performance for all the sites (except IVO). The specific functions for the optimal and generic scheme of environmental modifiers are provided in Table S4.


Figure 8 illustrates the uncertainty associated with the model representations of environmental influences on heterotrophic respiration. Most simulations within the grey area (corresponding to the grey dots within the good-performance boxes in Figure 7) employed the modified ELMV1-ECA moisture scalar and the $Q_{10}$-based temperature scalar, differing from each other by using different parameter values (e.g., $Q_{10}$, $Sf_{op}$, and $b$). At ATQ, the site with the thicker active layer, results from

simulations using moisture-dependency functions documented in Sierra et al. (2015) (Table S2 and Figure S1) were notably different than those using the moisture scalar of ELMv1-ECA. For the Sierra et al. (2015) empirical moisture functions, the influence of liquid moisture content on heterotrophic respiration is uniformly applied to all active soil layers, even though the soil properties are quite different vertically. ELMv1-ECA's moisture scalars (including the original scheme), in contrast,





reasonably explained the varying influence along the vertical soil profile. Thus, the simulations with moisture functions documented in Sierra et al. (2015) (i.e., different than the improved ELMV1-ECA moisture scalar) generally overestimated $CO_2$ and $CH_4$ emissions, especially during the warm season when the thaw depth is deep and soil wetness is high, thus permitting large moisture modifier scalar applied to the base decomposition rates.

Both the optimal and the generic decomposition scheme used the modified ELMv1-ECA moisture scalar (see Table S4),
which assigns small thresholds for the moisture scalar and also incorporates oxygen stress when soil wetness exceeds a favourable threshold (0.65 here) for decomposition. Imposing small thresholds for moisture scalar effectively prevents the possibility of zero respiration in subfreezing soils during wintertime. This change exerts more impact on cold sites, such as the two Barrow sites, due to the smaller supercooled liquid water under the colder temperature. Thus, the improved NSEs for $CO_2$ and $CH_4$ emissions at BES/CMDL and BEO were larger than those at ATQ (Figure 7; yellow or magenta vs. blue).
Since the temperature at ATQ was not cold enough to make the supercooled liquid water content small enough to give a zero moisture scalar, the microbial respiration was not completely shut down with the original decomposition modifier at this site. Indeed, at ATQ, where cold-season temperatures are relatively warmer than at BES/CMDL and BEO, simulations with the original ELMv1-ECA environmental modifier (i.e., "NewPC_OriDecom_NewCH4"; discussed in Section 3.1.2), already released much more $CO_2$ and $CH_4$ throughout the cold season than in the baseline simulations, owning to the improved
simulations of soil temperature and moisture, and the modifications for $CH_4$ transport.

The $Q_{10}$-based temperature functions mediate the response of microbial respiration more over the warm season than the cold season due to the larger sensitivity of heterotrophic respiration to warm temperatures than to subfreezing temperatures (see Figure S1d). The different SOC decomposition $Q_{10}$ values employed directly impact soil HR and thus $CH_4$ and $CO_2$
emissions, and also indirectly impact vegetation nutrient assimilation and thus primary productivity (Figure 2). Vegetation growth, on the other hand, impacts $CH_4$ emissions because the $CH_4$ component transported to the surface via vegetation aerenchyma tissue generally dominates the total emissions and thus determines the seasonal peak and general seasonality of $CH_4$ emissions. When temperature is below the reference temperature (i.e., $T_{ref}$, here is 25°C), a smaller $Q_{10}$ permits larger HR and produces more $CH_4$ and $CO_2$, increases warm-season $CO_2$ uptake via photosynthesis; and increases belowground
biomass and aerenchyma tissue and thereby $CH_4$ transport to the atmosphere. Thus, the seasonality of $CH_4$ and $CO_2$ net emissions are closely linked through vegetation primary productivity, which vary from site to site. For cold sites (i.e., BES/CMDL and BEO), the sensitivity of simulated $CH_4$ to $Q_{10}$ values is larger than the sensitivity of $CO_2$ net flux to $Q_{10}$ because cold temperature suppresses vegetation growth (i.e., $CO_2$ uptake); while for the warm site (i.e., ATQ), both $CH_4$ and $CO_2$ net flux are very sensitive to the $Q_{10}$ values. Summarizing, the cold sites (i.e., BES/CMDL and BEO) better match $CO_2$
and $CH_4$ emissions observations with smaller $Q_{10}$ values (1.7 or 1.8) than for the warmer site (i.e., 2.1 for ATQ; Table S4). The generic decomposition scheme used a $Q_{10}$ value of 2.0, which provided the best overall performance at all three sites (Table S4).



The extended ZCPs, the revised environmental modifier to decomposition, and the modified cold-season $CH_4$ transport
mechanism, together resulted in the largest improvements for both $CO_2$ and $CH_4$ emissions, especially over the cold season.
In the next section, we quantify the cold season contribution of $CO_2$ and $CH_4$ emissions and then estimate the historical
trends of seasonal $CO_2$ and $CH_4$ emissions from 1950 to 2017.

**4.3 Cold-season Contribution of $CH_4$ and $CO_2$ net emissions and Historical Trends**

To better verify the cold-season contribution of $CH_4$ and $CO_2$ emissions to the annual budget, a multi-year average approach
was taken because of discontinuity in the observed time series. The new simulation results with the optimal decomposition
scheme (yellow; Figure 9) showed greatly enhanced performance at three of the study sites in terms of capturing the
averaged seasonal cycle, especially for the cold-season months (Sep. to May; Figure 9), reducing site-averaged MAEs in
cold-season total $CH_4$ and $CO_2$ emissions by 84% and 81% , respectively. Specifically, compared to baseline results, the new
simulation results showed 0.94 gC m$^{-2}$ and 55.6 gC m$^{-2}$ increases in site-averaged cold-season $CH_4$ and $CO_2$ emissions,
respectively. The observed cold-season $CH_4$ emissions contributed at least ~40% to the annual total at three of the study
sites, of which about half occurred in September and October (Figure 10, Table 5), i.e., the two months hosting the major
part of ZCPs for the top to intermediate soil layers. The simulated percentage of cold-season contributions to the annual
totals were close to observed values, i.e., 38%, 41%, 28% vs. 45%, 42%, 45% for BES/CMDL, BEO, and ATQ,
respectively. The simulated contribution of early cold season (Sep. and Oct.) $CH_4$ emissions to the cold-season total was
62%, 52%, and 60% for the three sites, in comparison with the observed 47%, 58%, and 43%, showing slightly
overestimations.

The new simulations accurately captured the observed cold-season contributions of both $CH_4$ and $CO_2$ emissions (Table 5),
and the model improvements were larger for cold sites (i.e., BES/CMDL and BEO) than for the warmer site (i.e., ATQ), as
discussed above. Specifically, at ATQ, despite the small biases in the annual total $CH_4$ emission (i.e., -0.16 gC m$^{-2}$) and the
early cold season component (i.e., -0.05 gC m$^{-2}$), the new simulation underestimated the cold-season proportion of annual
emissions, i.e., simulated 28% vs. observed 45%. In contrast, biases in contribution percentages were only 2% and 7% at
BES/CMDL, and 3% and 1% at BEO for the early cold season and cold-season period, respectively. The updated ELMv1-
ECA also provided improved cold-season $CO_2$ emissions, showing small biases of -2.44 gC m$^{-2}$ (3% of the observation) and
-1.5 gC m$^{-2}$ (2% of the observation) for BES/CMDL and BEO, respectively. Compared to BES/CMDL and BEO, results for
ATQ showed relatively larger bias of -23.9 gC m$^{-2}$ (41% of the observation). The observed multi-year averaged annual $CO_2$
net flux was 19.9 gC m$^{-2}$ (source), 31.8 gC m$^{-2}$ (source), and -3.8 gC m$^{-2}$ (sink) at BES/CMDL, BEO, and ATQ, respectively.
However, due to the large discontinuity in $CO_2$ observations, especially over the warm season (Figure 8), the calculated
annual $CO_2$ budget is uncertain. Still, we can characterize the $CO_2$ budget with simulated results using the updated ELMv1-



ECA. We find that the simulated cold-season $CO_2$ emissions were larger than the warm-season $CO_2$ net uptake at all three sites (Figure 10, Table 5). The released $CO_2$ over the early cold season (September and October) accounted for 54%, 50%, and 72% of the total emissions throughout the cold season for BES/CMDL, BEO, and ATQ, respectively.

Through trend analysis between 1950 and 2017, we found that the ZCP durations showed increasing trends at all three sites, with ZCP trends increasing with depth (Table 6). At ATQ, a warmer site than BES/CMDL and BEO, the trends of ZCP durations increase from 0.14 to 0.49 days $yr^{-1}$ along the vertical soil profile. The $CO_2$ emissions during the 12 cm ZCP and during cold-season months (September to May) both showed increasing trends at all three sites (Table 7), ranging from 0.19 to 0.26 gC $m^{-2}$ $yr^{-1}$ for the 12 cm ZCP, and from 0.33 to 0.38 gC $m^{-2}$ $yr^{-1}$ for the entire cold season period. The annual $CO_2$
net flux showed positive trends, but they were not statistically significant. Annual $CH_4$ emissions showed an increasing trend at ATQ with a rate of 10.6 mgC $m^{-2}$ $yr^{-1}$, but not at the two Barrow sites; cold-season $CH_4$ emissions did not show significant trends at all the sites. In a companion paper, we discuss the regional trends of the spatially averaged $CO_2$ emissions simulated by the updated ELMv1-ECA with the identified generic decomposition scheme.

## 5 Summary and Discussion

In this study, we improved ELMv1-ECA simulated subsurface soil temperature, zero-curtain period durations, and cold-season $CH_4$ and $CO_2$ net emissions at Alaskan North Slope tundra sites. We first improved the numerical representation of coupled water and heat transport with freeze-thaw processes via modifying ELMv1-ECA's phase-change scheme. Then, we revised the dependency of soil decomposition rates on soil temperature and moisture. We further refined the cold-season methane processes by updating upper boundary resistance that allows $CH_4$ to be emitted from frozen soils through snow to
the atmosphere. We also used the updated ELMV1-ECA to estimate historical trends of cold-season $CH_4$ and $CO_2$ net emissions at the Alaskan tundra sites from 1950 to 2017. This study is among the first efforts toward improving simulations of zero-curtain periods and cold-season carbon emissions over Arctic tundra by ESMs. The strategy of improving ELMV1-ECA phase-change scheme and environmental controls on microbial activity can be easily applied to other global land models.


With the revised phase-change scheme, the updated ELMv1-ECA greatly improved site-scale simulations of soil temperature, soil moisture, and zero-curtain period. Specifically, the RMSE of daily subsurface soil temperature was substantially reduced compared to the baseline simulation, showing site-averaged improvements ranging from 58% to 87% over the early cold season (Sep. to Oct.) and from 36% to 46% over the annual cycle for soil layers within the active layer.
The evaluation of simulated liquid water content with the new phase-change scheme, although limited by availability of observations, showed a relative reduction in RMSE as high as 43% for the 5th layer at ATQ, and site-averaged improvements of 15% and 21% for the 4th and 5th layer, respectively. Simulated ZCP durations were also greatly improved, with, e.g.,



relative reductions in MAEs of 65%, 65%, 66%, and 50% for the 4th layer (about 12 cm) at BES/CMDL, BEO, ATQ, and IVO, respectively.


Based upon the improved simulations of soil temperature and moisture with the new phase-change scheme, the identified an optimal SOC decomposition scheme, and the revised methane module, the site-averaged mean annual errors of cold-season emissions were reduced by 84% and 81% for $CH_4$ and $CO_2$, respectively. We also found that $CH_4$ and $CO_2$ emissions over the early cold season, i.e., September and October, which usually accounts for most of the zero-curtain period,

contributed more than 50% of the total emissions throughout the cold season (September to May). Zero-curtain period durations showed increasing trends from 1950 to 2017, with larger trends in deeper soil layers. Although the annual $CO_2$ emissions did not show statistically significant trends, both $CO_2$ emissions during the 12 cm depth zero-curtain period and the entire cold-season period (Sep. to May) showed increasing trends.

Although showing improvements compared to baseline results, the new simulations generally overestimated the contribution of early cold season $CO_2$ emissions. Many reasons could contribute to the overestimations, including poor representation of coupled biogeochemical and hydrological processes in the localized permafrost soil environment, the lack of accurate representation of inundated hydro-ecological dynamics, underestimation of snow accumulation due to micro-topographic effects and thus the snow insulation to the ground (e.g., Bisht et al., 2018), among others. Strong microtopographic impacts

on $CO_2$ and $CH_4$ emissions across seven landscape types in Barrow, Alaska, were recently reported (Wang et al. (2019); Grant et al., 2017a; Grant et al., 2017b). In addition, the single static multiplicative function ($f_{total} = f_T f_W f_O f_D$) used to parameterize the total impact of environmental conditions on respiration might not be appropriate (Tang and Riley, 2019). Also, inappropriately prescribed land cover at the site scale or inaccurate climate forcing (particularly air temperature and precipitation; Chang et al. (2019)) could all impact snow accumulation processes (Tao et al., 2017), which can significantly

impact $CO_2$ and $CH_4$ emission simulations. Customizing the complex local ecosystem vegetation community might be feasible at the site scale, however, it is less possible for regional or global land model simulations. This issue calls for the importance of upscaling methods to model (e.g., Pau et al., 2016; Liu et al., 2016) and measure (e.g., Natali et al., 2019; Virkkala et al., 2019) carbon and water cycle dynamics at the regional and global scales.

Given the persistent warming and the continued more severe warming in the cold season (Box et al., 2019), we envision continuing increases in cold-season $CO_2$ and $CH_4$ emissions from the permafrost tundra ecosystem. The increasing rate of cold-season heterotrophic respiration (releasing $CO_2$) may become larger than the trend of warm-season vegetation $CO_2$ uptake under future climate. To accurately characterize cold-season emissions and warm-season net uptake, models have to correctly simulate both components, which, however, few models can do. The updated ELMv1-ECA, with the enhanced

capacity to reproduce cold-season $CO_2$ and $CH_4$ emissions proven by this study, can serve as a starting point to better predict permafrost carbon responses to future climate. Finally, the complex water-carbon interactions require modelling systems



with fully coupled hydrological-thermal-biogeochemical processes to better predict the carbon budget in permafrost regions under future climate.

## Code/Data availability

The observations used in this study are available at http://dx.doi.org/10.3334/ornldaac/1300 and https://doi.org/10.3334/ornldaac/1562. The UAF observations are available at http://permafrost.gi.alaska.edu. The updated version of ELMv1-ECA will be available at GitHub.

## Author contributions

JT assembled observations, developed the methodology, conducted model simulations, analysed results, and wrote and
revised the paper. QZ contributed to experiment design, editing the original and revised manuscript. WJR and RBN edited the original and revised manuscript, and provided valuable discussion and guidance.

## Competing interests

The authors declare that they have no conflict of interest.

## Acknowledgments

We are grateful for valuable discussions with Jinyun Tang, Roisin Commane, Xiyan Xu, Kai Yang, and Chenghai Wang. We thank the anonymous reviewers for their helpful comments. This material is based upon work supported by the U.S. Department of Energy, Office of Science, Office of Biological and Environmental Research under Award Number DE-SC0019063.

## Appendices: Description of Relevant Modules within ELMv1-ECA

Here we describe the heat transfer in subsurface soils, the carbon decomposition, and the methane module within the ELMv1-ECA that are of particular relevance to our study.

### Appendix A Subsurface Heat Transfer

ELMv1-ECA approximates the subsurface heat transfer process with a one-dimensional heat diffusion equation:





$$c\frac{\partial T}{\partial t} = \frac{\partial}{\partial z}\left(\lambda\frac{\partial T}{\partial z}\right),$$ (Eq. A1)

where $T$ is the soil temperature (K), $c$ is the volumetric soil heat capacity (J m$^{-3}$ K$^{-1}$), $\lambda$ is soil thermal conductivity (W m$^{-1}$ K$^{-1}$), and $z$ is the soil depth (m) of the ELMv1-ECA soil layers. The ELMv1-ECA soil column consists of 15 layers, with soil thickness increasing exponentially with depth. The bottom of soil column is down to 42 m, and the top 10 layers are hydrologically active with layer node depth as 0.0071 m, 0.0279 m, 0.0623 m, 0.1189 m, 0.2122 m, 0.3661 m, 0.6198 m, 1.0380 m, 1.7276 m, 2.8646 m, respectively. The soil heat capacity and thermal conductivity is updated at each time step based on the fractions of soil matrix components, i.e., liquid water content, ice content, and soil solids. The impact of organic carbon on soil thermal and hydraulic properties was incorporated as a linear combination of the counterparts properties of mineral soil and organic matter (Lawrence and Slater, 2008). To solve the (Eq. A1), ELMv1-ECA employs the Crank-Nicholson method, resulting in a tridiagonal system equation. We assume a zero-flux bottom boundary condition. The top boundary condition is estimated by solving the energy balance equation at the air and ground interface, with additionally an overlying five-layer snow model and a one-layer surface water model in between. When snow and surface water present, ELMv1-ECA incorporates the snow layers and surface water layer into the tridiagonal system to solve the heat transfer along the entire column.

ELMv1-ECA incorporates freeze-thaw processes of soil water in a decoupled way. Specifically, the model determines the onset of melting or freezing by soil temperature initially solved at time step $n + 1$ without consideration of the phase change process, denoted as $T_i^{n+1}$, i.e.,

$$
\begin{aligned}
T_i^{n+1} > \ T_f \ and \ w_{ice,i}^n > 0 & \qquad \text{melting} \\
T_i^{n+1} < \ T_f \ and \ w_{liq,i}^n > w_{liq,max,i}^{n+1} & \qquad \text{freezing}
\end{aligned}
\ , 
$$ (Eq. A2)

where $T_f$ is the freezing temperature of water (0°C in Kelvin, i.e., 273.15 K), $w_{ice,i}^n$ and $w_{liq,i}^n$ is the mass of ice and liquid water (kg m$^{-2}$) of layer $i$ , and $w_{liq,max,i}^{n+1}$ (kg m$^{-2}$) is the supercooled liquid water that is allowed to coexist with ice given the subfreezing soil temperature $T_i^{n+1}$. This $w_{liq,max,i}^{n+1}$ varies with soil texture and temperature and is calculated by the freezing point depression equation (Niu and Yang, 2006),

$$w_{liq,max,i}^{n+1} = \Delta z_i \theta_{sat,i}\left[\frac{10^3 L_f\left(T_f - T_i^{n+1*}\right)}{g T_i^{n+1*}\psi_{sat,i}}\right]^{-1/B_i} \ ,$$ (Eq. A3)

where $\Delta z_i$ is the soil thickness of the $i$th layer (in mm), $\theta_{sat,i}$ represents the soil porosity (i.e., the saturated volumetric water content), $L_f$ is the latent heat of fusion (J kg$^{-1}$), $B_i$ is the Clapp and Hornberger exponent (Clapp and Hornberger, 1978), g is the gravitational acceleration (m s$^{-2}$), and $\psi_{sat,i}$ is the soil texture-dependent saturated matric potential (mm).





The rate of phase change is initially assessed from the heat excess (or deficit) needed to change the estimated temperature to

the freezing point. Specifically, the model first computes the energy ($H_i$) needed for adjusting current soil temperature ($T_i^{n+1}$) to $T_f$:

$$H_i = -c_i \frac{\Delta z_i}{\Delta t} T_{inc} + \left(1 - f_{sno} - f_{h2osfc}\right) \frac{\partial h}{\partial T} T_{inc} \quad i = 1$$

$$H_i = -c_i \frac{\Delta z_i}{\Delta t} T_{inc} \quad\quad\quad\quad\quad\quad\quad\quad i > 1$$

(Eq. A4)

where $T_{inc} = T_f - T_i^{n+1}$, $h$ is ground heat flux, $f_{sno}$ and $f_{h2osfc}$ is the snow and surface water fraction within the grid cell,

respectively. The mass change involved then is computed as $H_m = \frac{H_i \Delta t}{L_f}$ (i.e., $-c_i \frac{\Delta z_i}{L_f} T_{inc}$ for soils below the top interface

layer). That is, the mass of ice increased/decreased by freezing/melting is $-H_m$, releasing/absorbing energy $H_i$ to bring

up/down the current soil temperature to $T_f$. Accordingly, the ice and liquid mass are adjusted as:

$$w_{ice,i}^{n+1} = \begin{cases} \min\left(w_{ice,i}^n + w_{liq,i}^n - w_{liq,max,i}^{n+1*}, \ w_{ice,i}^n - H_m\right) & w_{liq,i}^n + w_{ice,i}^n \geq w_{liq,max,i}^{n+1*} \\ 0 & w_{liq,i}^n + w_{ice,i}^n < w_{liq,max,i}^{n+1*} \end{cases}$$

(Eq. A5)

$$w_{liq,i}^{n+1} = \max\left(w_{liq,i}^n + w_{ice,i}^n - w_{ice,i}^{n+1}, 0\right)$$

The $H_i$ then is adjusted to $H_{i*}$, calculated as $H_{i*} = H_i - \frac{L_f\left(w_{ice,i}^n - w_{ice,i}^{n+1}\right)}{\Delta t}$. The $H_{i*}$ then is the ultimately determined latent heat

and is used to further readjust soil temperature as in equation (Eq. A6),

$$T_i^{n+1*} = T_f + \frac{\Delta t}{c_i \Delta z_i} H_{i*} = T_f - \frac{L_f\left(w_{ice,i}^n - w_{ice,i}^{n+1}\right)}{c_i \Delta z_i},$$

(Eq. A6)

in which the temperature adjusted to $T_f$ is further increased by $-\frac{L_f\left(w_{ice,i}^n - w_{ice,i}^{n+1}\right)}{c_i \Delta z_i}$ due to soil freezing since $w_{ice,i}^{n+1} \geq w_{ice,i}^n$, or

decreased due to melting when $w_{ice,i}^{n+1} < w_{ice,i}^n$.

We revised the phase-change scheme mainly through incorporating a phase-change efficiency ($\varepsilon$) and replacing the constant

freezing point $T_f$ with the temperature of the freezing point depression in (Eq. A2). The phase-change efficiency, introduced

by Le Moigne et al. (2012) and adopted by Masson et al. (2013) and Yang et al. (2018a), is calculated as,

$$\begin{cases} \varepsilon_{liq,i}^n = \frac{\theta_{liq,i}^n}{\theta_{sat,i}} & for\ freezing \\ \varepsilon_{ice,i}^n = \frac{\theta_{ice,i}^n}{\theta_{sat,i}} & for\ melting \end{cases},$$

(Eq. A7)

where $\theta_{liq,i}^n$ and $\theta_{ice,i}^n$ is the soil liquid and ice volumetric water content of layer $i$ at previous time step $n$, respectively, and

$\theta_{sat,i}$ represents the soil porosity (i.e., the saturated volumetric water content).





The temperature of the freezing point depression, as a virtual temperature ($Tv$) reversely derived from the freezing point temperature-depression equation, i.e., $\psi(T) = \frac{L_f(T_f - T_i)}{10^3 T}$ (Fuchs et al., 1978; Cary and Mayland, 1972), is calculated as,

$$Tv_i^{n+1*} = \frac{10^3 L_f T_f}{10^3 L_f + g\psi_i^n} \quad , \tag{Eq. A8}$$

where $L_f$ is the latent heat of fusion (J kg$^{-1}$) and g is the gravitational acceleration (m s$^{-2}$). $\psi_i^n$ is the soil water potential

(mm), calculated as the soil water retention curve of Clapp and Hornberger (1978), i.e., $\psi_i^n = \psi_{sat,i} \left( \frac{\theta_{liq,i}^n}{\theta_{sat,i}} \right)^{-B_i}$, where

$\theta_{liq,i}^n = w_{liq,i}^n / \Delta z_i$ as in (Eq. A3), $B_i$ is the Clapp and Hornberger exponent, and $\psi_{sat,i}$ is the soil texture-dependent saturated matric potential (mm).

**Appendix B Decomposition Cascade Model**

Within the ELMv1-ECA Century decomposition cascade model, the respiration fractions are parameterized as the fraction of the decomposition carbon flux out of each carbon pool, including litter and soil organic matter. The base decomposition rate is modified by a function representing environmental controls on soil decomposition which accounts for the impacts of individual factors including temperature ($f_T$) and moisture ($f_W$), an oxygen scalar ($f_O$), and a depth scalar ($f_D$), in a multiplicative way, i.e., $f_{total} = f_T f_W f_O f_D$.


We use a Q$_{10}$-based standard exponential function to account for the temperature effect on decomposition,

$$f_T = Q_{10}^{\left(\frac{T - T_{ref}}{10}\right)}, \tag{Eq. B9}$$

where Q$_{10}$ = 1.5 on default, which is consistent with ecosystem-level observations (Mahecha et al., 2010), and $T_{ref}$ is the reference temperature (25°C). During cold seasons when soil temperature becomes subfreezing, respiration continues but with more controls from liquid water stress. The original moisture scalar ($f_W$) within ELMV1-ECA is given in the

formulation, calculated as,

$$f_W = \begin{cases} 0 & For \ \psi_i < \psi_{min} \\ \dfrac{log(\psi_{min}/\psi_i)}{log(\psi_{min}/\psi_{max})} & For \ \psi_{min} \leq \psi_i \leq \psi_{max} \\ 1 & For \ \psi_i > \psi_{max} \end{cases} \quad , \tag{Eq. B10}$$

where $\psi_i = \psi_{max} \left( \frac{\theta_{liq,i}}{\theta_{sat,i}} \right)^{-B_i}$ is the soil water potential, where $B_i$ is the Clapp and Hornberger exponent (Clapp and Hornberger, 1978). In frozen soil, the soil water potential is related to soil temperature through the freezing point depression





equation, i.e., $\psi_i = \frac{L_f(T_f - T_i)}{10^3 T}$ (Fuchs et al., 1978; Cary and Mayland, 1972) in the supercooled water formulation (Niu and

Yang, 2006). Thus, the liquid water stress on decomposition is translated into dependency on temperature when soil

temperature is below the freezing point.

ELMv1-ECA approximates oxygen stress ($f_O$) as the ratio of available oxygen to that demanded by decomposers, and has a

minimum value of 0.2 (Oleson et al., 2013). As described by section 3.1.2, we now incorporate the oxygen stress into the

moisture scalar. The revised moisture scalar $f_W{}^*$ is calculated as below,

$$f_W{}^* = \begin{cases} max\left[\dfrac{log\left(\dfrac{\psi_{min}}{\psi_{max}Sf_{liq}} - B_i\right)}{log\left(\dfrac{\psi_{min}}{\psi_{max}}\right)}, f_{W\_min}\right] & For\ Sf_{liq} < Sf_{op} \\[4ex] max\left[\dfrac{log\left(\dfrac{\psi_{min}}{\psi_{max}Sf_{liq}} - B_i\right)}{log\left(\dfrac{\psi_{min}}{\psi_{max}}\right)} \times \left(\dfrac{1 - Sf_{liq}}{1 - Sf_{op}}\right)^b, f_{W\_min}\right] & For\ Sf_{liq} \geq Sf_{op} \end{cases}, \qquad \text{(Eq. B11)}$$

where $Sf_{liq}$ is the degree of saturation, calculated as the ratio of soil volumetric liquid water content to porosity $\left(i.e., \frac{\theta_{liq,i}}{\theta_{sat,i}}\right)$,

$Sf_{op}$ is an optimal threshold beyond which the decomposition will be suppressed by oxygen stress, and b is a parameter

controlling the shape of the decreasing limb, and $f_{W\_min}$ is a minimum threshold for $f_W{}^*$.

The depth scalar ($f_D = exp\left(-\frac{z_i}{Z_\tau}\right)$) represents unresolved other depth-dependent processes (e.g., soil microbial dynamics,

priming effects, etc.) (Koven et al., 2013b; Lawrence et al., 2015; Koven et al., 2015). Applying the depth scalar to

decomposition rates would exponentially decrease the respiration fluxes along with the vertical soil layers. The $Z_\tau$ is the e-

folding depth for decomposition, and by default $Z_\tau$ is 0.5 m (Oleson et al., 2013).

**Appendix C Methane Model**

The ELMv1-ECA methane model includes the representations of $CH_4$ production, oxidation, and three pathways of transport

(i.e., aerenchyma tissues, ebullition, aqueous and gaseous diffusion), and solves the transient reaction diffusion equation for

$CH_4$. ELMv1-ECA estimates $CH_4$ production ($P$; mol m$^{-3}$ s$^{-1}$) in the anaerobic portion of the soil column based on the

upland heterotrophic respiration (HR; mol C m$^{-2}$ s$^{-1}$) from soil and litter, further adjusted by factors representing influence

from soil temperature ($f_T$), pH ($f_{pH}$), redox potential ($f_{pE}$), and seasonal inundation condition ($S$) (Riley et al., 2011),

expressed as,

$P = HR \times f_{CH_4} \times f_T \times f_{pH} \times f_{pE} \times S.$ \qquad (Eq. C12)



The $f_{CH_4}$ is a fraction of anaerobically mineralized carbon atoms becoming $CH_4$. Detailed explanation on other factors can be found in (Riley et al., 2011). The methane production $P$ will be directly impacted by the changes to water and heat transfer (Appendix A) and HR (Appendix B). The ultimately estimated $CH_4$ emissions are also controlled by oxidation, transport mechanisms (i.e., aerenchyma transport, ebullition, and diffusion), and the upper boundary resistance. Detailed descriptions

on $CH_4$ oxidation and transport mechanisms are provided in (Riley et al., 2011). Here we modified $CH_4$ transport mechanisms for facilitating reasonable cold-season $CH_4$ emissions.

Vascular plants aerenchyma tissues serve as diffusive pathways for $CH_4$ to transport from soil layer $z$ ($A(z)$, mol m$^{-2}$ s$^{-1}$) to the atmosphere, calculated as:

$$A(z) = (C(z) - C_a)/\left(\frac{r_L z}{Dp T_{aere} \rho_r(z)} + r_a\right) \quad , \tag{Eq. C13}$$

where $C(z)$ and $C_a$ is the gaseous $CH_4$ concentration (mol m$^{-3}$) in soil depth $z$ and in the atmosphere, respectively; $r_a$ is the aerodynamic resistance (s m$^{-1}$); $D$ is the gas diffusion coefficient (m$^2$ s$^{-1}$); $p$ is aerenchyma porosity (-); $r_L$ is the ratio of root length to vertical depth (i.e., root obliquity); and $\rho_r(z)$ is the root fraction in soil depth $z$ (-). $T_{aere}$ is the specific aerenchyma area (m$^2$ m$^{-2}$), and is expressed as,

$$T_{aere} = \frac{f_N N_a LAI}{0.22}\pi R^2, \tag{Eq. C14}$$

where $R$ represents the aerenchyma radius (=2.9×10$^{-3}$ m); $N_a$ is the annual net primary production (NPP), and $f_N$ is the

belowground fraction of current NPP; and the factor 0.22 is the amount of carbon per tiller. We integrate the emissions from ice cracks and remnants of aerenchyma tissues with (Eq. C14) by removing temperature limitation and applying a small $T_{aere}$ during winter time, where $T_{aere}$ now represents areas adding up ice crack fractions and remnants of aerenchyma tissues.

ELMv1-ECA estimates aqueous diffusion below water table as,

$$D_e = \begin{cases} D_0 \theta_{sat}^2 & For\ T \geq 0°C \\ D_0 \theta_{sat}^2 f_{frzsoil} & For\ T < 0°C \end{cases}, \tag{Eq. C15}$$

where $D_0$ is the diffusion coefficient (m$^2$ s$^{-1}$), $\theta_{sat}$ is the soil porosity, $f_{frzsoil}$ is a scaling factor for frozen soils, defined as $f_{frzsoil} = \frac{V_{liq}}{V_{liq}+V_{ice}}$ where $V_{liq}$ and $V_{ice}$ is the volume (m$^3$ m$^{-3}$) of liquid water and ice, respectively. In subfreezing soils when $T < 0°$, $D_e$ is largely limited by liquid water content. Upon sensitivity experiments, we alleviated this limitation by assuming a half deduction for the diffusion coefficient in saturated, frozen soils, i.e., $f_{frzsoil}$ = 0.5. We also decreased snow resistance

by introducing new scale factors (Table 2) which intend to increase the conductance at the upper boundary when snow presents.



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



**Table 1: Specific modifications made to ELMv1-ECA. Process interactions are illustrated in Figure 2.**

| | Part 1 – Phase-change scheme within the heat transfer module | | Part 2 – Environmental modifier to the base decomposition rate | | Part 3 – Methane module | |
|---|---|---|---|---|---|---|
| Relevant processes influenced | Water and heat transfer, plant and soil respiration, plant productivity, $CO_2$ fluxes and $CH_4$ emissions. | | Plant and soil respiration, plant productivity, $CO_2$ fluxes and $CH_4$ emissions. | | $CH_4$ emissions | |
| | Original | New | Original | New | Original | New |
| Variables or equations influenced | Eq. A2-6 | Imposing Eq. A7 and Eq. A8 to Eq. A2-A6 | Eq. B9-B10 | Eq. B11 and changes in Table S2 | Eq. C13-C15 | 1. Applying a minimum LAI to (Eq. C14) to mimic ice cracks and remnants of aerenchyma tissues in frozen soils, and permitting transport even when temperature is below 0°C. 2. Introducing new scale factors for snow resistance: scale_factor_gasdiff_snow = scale_factor_gasdiff*100 scale_factor_liqdiff_snow = scale_factor_liqdiff*100 3. Introducing a new scale factor for diffusivity in frozen soil: $f_{frzsoil}$= 0.5 in (Eq. C15) |




**Table 2: List of Designed Site-Scale Experiments. Process interactions among the three parts are illustrated in Figure 2.**

| Experiment Name | Part 1 - Phase Change Scheme within Heat Transfer Model | | Part 2 – Environmental Modifier within Carbon Decomposition Model | | Part 3 – Methane Model | |
|---|---|---|---|---|---|---|
| | Original | New | Original | New | Original | New |
| OriPC_OriDecom_OriCH4 (Baseline) | √ | | √ | | √ | |
| NewPC_OriDecom_OriCH4 | | √ | √ | | √ | |
| NewPC_OriDecom_NewCH4 | | √ | √ | | | √ |
| NewPC_NewDecom_NewCH4* (NewPC_OptimalDecom_NewCH4)# (NewPC_GenericDecom_NewCH4)$ | | √ | | √ | | √ |

*Replacing the original temperature- and moisture-dependency functions on decomposition rates with 200 new functions of environmental modifiers as listed in Table S2 in the supplementary file.

# "NewPC_OptimalDecom_NewCH4" is the optimal simulation among the 200 "NewPC_NewDecom_NewCH4" cases at each site.

$"NewPC_GenericDecom_NewCH4" means the simulation with the identified generic scheme that can be applied to regional simulation. The generic scheme is the common satisfactory scheme that provides the best overall performance for all the sites.






**Table 3: Mean absolute error (MAE) of simulated ZCP (days) with the original phase-change scheme (Ori_PC) and newly resized phase-change scheme (NewPC), and the relative improvement (%) of using the new phase-change scheme compared to the baseline results, calculated as 100% × (MAE_ZCP_OriPC – MAE_ZCP_NewPC) / MAE_ZCP_OriPC.**

| | BES&CMDL | | | BEO | | | ATQ | | | IVO | | |
|---|---|---|---|---|---|---|---|---|---|---|---|---|
| | MAE_ZCP_OriPC (days) | MAE_ZCP_NewPC (days) | Improvement (%) | MAE_ZCP_OriPC (days) | MAE_ZCP_NewPC (days) | Improvement (%) | MAE_ZCP_OriPC (days) | MAE_ZCP_NewPC (days) | Improvement (%) | MAE_ZCP_OriPC (days) | MAE_ZCP_NewPC (days) | Improvement (%) |
| **Layer 1** | 38.80 | 31.40 | 19.07 | 37.60 | 33.20 | 11.70 | 26.33 | 13.33 | 49.37 | 54.00 | 51.50 | 4.63 |
| **Layer 2** | 29.20 | 14.20 | 51.37 | 27.40 | 12.60 | 54.01 | 24.33 | 5.67 | 76.71 | 50.50 | 37.50 | 25.74 |
| **Layer 3** | 35.20 | 18.40 | 47.73 | 33.60 | 16.80 | 50.00 | 28.00 | 9.33 | 66.67 | 55.75 | 30.25 | 45.74 |
| **Layer 4** | 29.60 | 10.40 | 64.86 | 30.60 | 10.60 | 65.36 | 28.67 | 9.67 | 66.28 | 61.50 | 30.50 | 50.41 |
| **Layer 5** | 18.00 | 11.40 | 36.67 | 17.60 | 10.80 | 38.64 | 27.67 | 17.33 | 37.35 | 54.50 | 22.00 | 59.63 |
| **Layer 6** | 77.40 | 12.20 | 84.24 | 77.40 | 13.00 | 83.20 | 61.67 | 36.67 | 40.54 | 68.00 | 14.75 | 78.31 |
| **Layer 7** | NaN | NaN | NaN | NaN | NaN | NaN | NaN | NaN | NaN | 151.33 | 46.67 | 69.16 |






**Table 4: RMSE (°C) of simulated soil temperatures with the original phase-change (PC) scheme and newly resized PC scheme. NaN represents the cases when observations are not available.**

| Model Layer (Node Depth) | BES&CMDL | | BEO | | ATQ | | IVO | |
|---|---|---|---|---|---|---|---|---|
| | Ori_PC | New_PC | Ori_PC | New_PC | Ori_PC | New_PC | Ori_PC | New_PC |
| Layer 1 (0.01 m) | 5.66 | 3.82 | 5.45 | 3.85 | 6.47 | 3.77 | 9.12 | 5.42 |
| Layer 2 (0.03 m) | 5.36 | 3.35 | 5.16 | 3.45 | 6.42 | 3.66 | 9.08 | 5.22 |
| Layer 3 (0.06 m) | 5.32 | 3.16 | 5.16 | 3.28 | 6.38 | 3.54 | 8.87 | 4.91 |
| Layer 4 (0.12 m) | 5.25 | 2.92 | 5.22 | 3.00 | 6.33 | 3.40 | 8.87 | 4.81 |
| Layer 5 (0.21 m) | 5.15 | 2.72 | 4.90 | 2.82 | 6.24 | 3.32 | 8.76 | 4.60 |
| Layer 6 (0.37 m) | 4.70 | 2.56 | 4.70 | 2.56 | 6.15 | 3.50 | 8.67 | 4.42 |
| Layer 7 (0.62 m) | 4.41 | 2.33 | 4.41 | 2.34 | NaN | NaN | 8.38 | 4.08 |
| Layer 8 (1.04 m) | 4.23 | 2.13 | 4.22 | 2.14 | NaN | NaN | 7.75 | 3.46 |
| Layer 9 (1.73 m) | 4.33 | 2.04 | 4.32 | 2.07 | NaN | NaN | NaN | NaN |
| Layer 10 (2.86 m) | 4.28 | 2.19 | 4.27 | 2.22 | NaN | NaN | NaN | NaN |
| Layer 11 (4.74 m) | 3.96 | 2.11 | 3.96 | 2.13 | NaN | NaN | NaN | NaN |
| Layer 12 (7.83 m) | 2.92 | 1.51 | 2.92 | 1.52 | NaN | NaN | NaN | NaN |
| Layer 13 (12.93 m) | 2.77 | 0.74 | 2.78 | 0.78 | NaN | NaN | NaN | NaN |
| Layer 14 (21.33 m) | NaN | NaN | NaN | NaN | NaN | NaN | NaN | NaN |
| Layer 15 (35.18 m) | NaN | NaN | NaN | NaN | NaN | NaN | NaN | NaN |






**Table 5: Total CH₄ emissions and CO₂ net flux over three seasonal periods, including the early cold season, cold season, and the warm season. The "ELM_New" here means NewPC_OptimalDecom_NewCH4 (Table 2). The percentage of each seasonal total CH₄ emissions to the annual total is included in the brackets.**

| Total CH₄ Emissions (gC m⁻²) | BES/CMDL | | | BEO | | | ATQ | | |
|---|---|---|---|---|---|---|---|---|---|
| | Early Cold Season (Sep. and Oct.) | Cold Season (Sep. to May) | Warm Season (Jun. to Aug.) | Early Cold Season (Sep. and Oct.) | Cold Season (Sep. to May) | Warm Season (Jun. to Aug.) | Early Cold Season (Sep. and Oct.) | Cold Season (Sep. to May) | Warm Season (Jun. to Aug.) |
| ELM_Baseline | 0.08 (4.7%) | 0.08 (5.1%) | 1.53 (94.9%) | 0.09 (5.5%) | 0.10 (6.2%) | 1.54 (93.8%) | 0.16 (15.1%) | 0.16 (15.3%) | 0.89 (84.7%) |
| ELM_New | 0.73 (23.3%) | 1.19 (37.9%) | 1.95 (62.1%) | 0.77 (21.7%) | 1.46 (41.4%) | 2.07 (58.6%) | 0.31 (17.6%) | 0.51 (28.3%) | 1.23 (70.7%) |
| Observation | 0.63 (21.0%) | 1.32 (44.5%) | 1.65 (55.5%) | 0.83 (24.4%) | 1.43 (41.9%) | 1.97 (58.1%) | 0.36 (19.2%) | 0.85 (44.7%) | 1.04 (55.3%) |
| Total CO₂ Net Flux (gC m⁻²) | BES/CMDL | | | BEO | | | ATQ | | |
| | Early Cold Season (Sep. and Oct.) | Cold Season (Sep. to May) | Warm Season (Jun. to Aug.) | Early Cold Season (Sep. and Oct.) | Cold Season (Sep. to May) | Warm Season (Jun. to Aug.) | Early Cold Season (Sep. and Oct.) | Cold Season (Sep. to May) | Warm Season (Jun. to Aug.) |
| ELM_Baseline | 31.27 | 31.38 | 2.03 | 30.99 | 31.14 | 13.91 | 40.46 | 40.86 | -26.05 |
| ELM_New | 48.50 | 89.94 | -61.87 | 49.10 | 97.65 | -55.47 | 59.14 | 82.55 | -46.64 |
| Observation | 43.60 | 87.50 | -67.61 | 28.20 | 96.14 | -64.33 | 24.29 | 58.64 | -62.41 |






**Table 6: Historical trend of ZCP durations (days year$^{-1}$) for each soil layer from 1950 to 2017. (Trends with $p > 0.05$ are not statistically significant.)**

|  | BES/CMDL | | BEO | | ATQ | |
|---|---|---|---|---|---|---|
|  | Trend (days yr$^{-1}$) | p Value | Trend (days yr$^{-1}$) | p Value | Trend (days yr$^{-1}$) | p Value |
| ZCP Duration of Layer 1 | -0.02 | 0.73 | -0.02 | 0.73 | 0.07 | 0.40 |
| ZCP Duration of Layer 2 | 0.09 | 0.03 | 0.09 | 0.04 | 0.14 | 0.05 |
| ZCP Duration of Layer 3 | 0.10 | 0.03 | 0.12 | 0.01 | 0.15 | 0.05 |
| ZCP Duration of Layer 4 | 0.10 | 0.07 | 0.10 | 0.09 | 0.21 | 0.01 |
| ZCP Duration of Layer 5 | 0.11 | 0.10 | 0.11 | 0.09 | 0.23 | 0.02 |
| ZCP Duration of Layer 6 | 0.37 | 0.51 | 0.35 | 0.56 | 0.49 | 0.00 |




**Table 7: Historical trend (1950 - 2017) in site-scale heterotrophic respiration, CH₄ emission, and CO₂ flux during the ZCP duration at 12 cm (4th layer), cold-season months (Sep. - May), and the whole annual cycle (Sep. – Aug.). (Trends with p > 0.05 are not statistically significant.)**

| | BES/CMDL | | BEO | | ATQ | |
|---|---|---|---|---|---|---|
| | **Trend of Heterotrophic Respiration** | | | | | |
| | Trend (g C m$^{-2}$ yr$^{-1}$) | p Value | Trend (g C m$^{-2}$ yr$^{-1}$) | p Value | Trend (g C m$^{-2}$ yr$^{-1}$) | p Value |
| **ZCP duration at 12 cm** | 0.02 | 0.17 | 0.02 | 0.24 | 0.07 | 0.00 |
| **Cold Season (Sep.-May)** | 0.09 | 0.00 | 0.09 | 0.00 | 0.13 | 0.00 |
| **Annual (Sep.-Aug.)** | 0.21 | 0.00 | 0.18 | 0.00 | 0.30 | 0.00 |
| | **Trend of CH₄ Emission** | | | | | |
| | Trend (mg C m$^{-2}$ yr$^{-1}$) | p Value | Trend (mg C m$^{-2}$ yr$^{-1}$) | p Value | Trend (mg C m$^{-2}$ yr$^{-1}$) | p Value |
| **ZCP duration at 12 cm** | -7.61 | 0.01 | -7.89 | 0.00 | -0.66 | 0.73 |
| **Cold Season (Sep.-May)** | -2.54 | 0.22 | -3.19 | 0.13 | 2.50 | 0.13 |
| **Annual (Sep.-Aug.)** | -4.98 | 0.20 | -5.63 | 0.15 | 10.56 | 0.00 |
| | **Trend of CO₂ Net Emissions** | | | | | |
| | Trend (g C m$^{-2}$ yr$^{-1}$) | p Value | Trend (g C m$^{-2}$ yr$^{-1}$) | p Value | Trend (g C m$^{-2}$ yr$^{-1}$) | p Value |
| **ZCP duration at 12 cm** | 0.20 | 0.00 | 0.19 | 0.00 | 0.26 | 0.00 |
| **Cold Season (Sep.-May)** | 0.36 | 0.00 | 0.33 | 0.00 | 0.38 | 0.00 |
| **Annual (Sep.-Aug.)** | 0.08 | 0.68 | 0.10 | 0.64 | 0.18 | 0.47 |






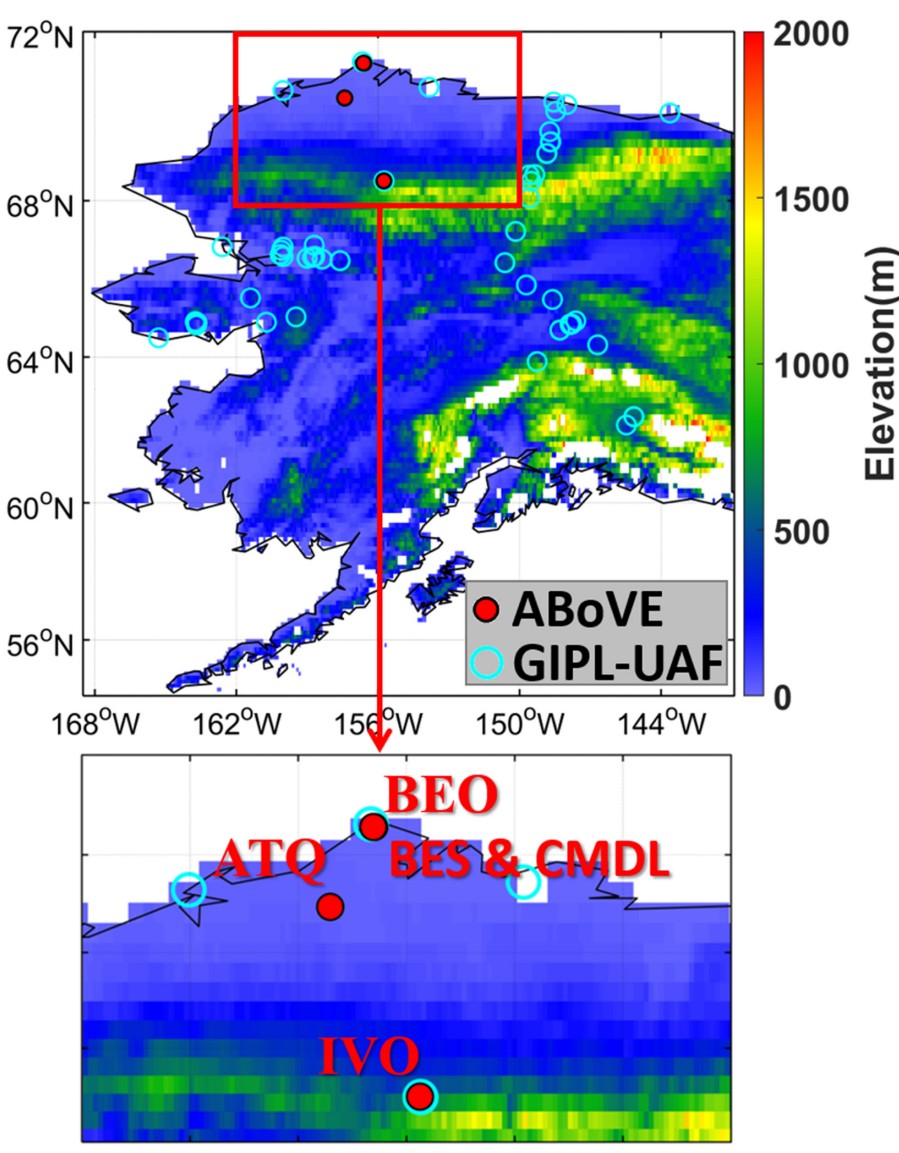

**Figure 1: Red dots indicate the five ABoVE flux tower sites used in this study. Cyan circles are GIPL-UAF permafrost sites.**





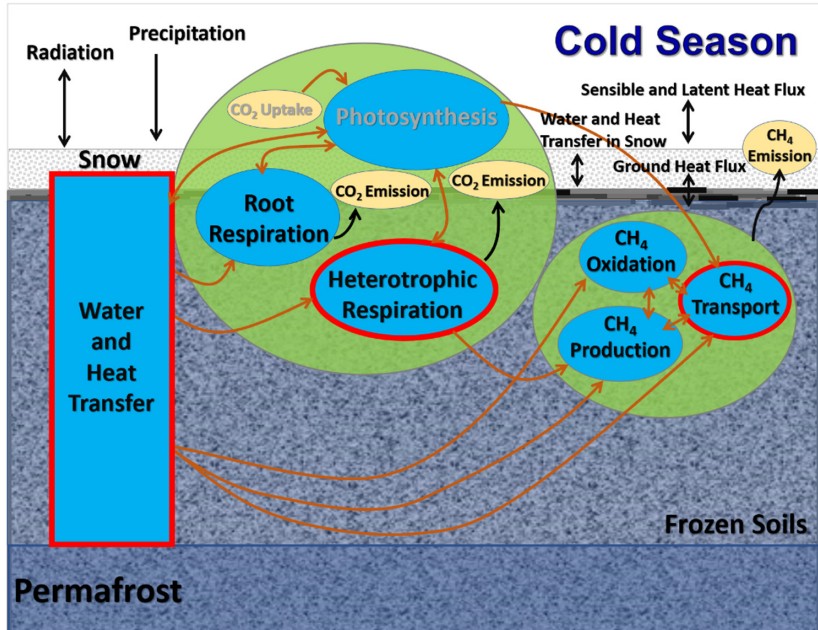


**Figure 2: Schematic diagram illustrating the interactions between the water and heat transfer module, vegetation and carbon decomposition module, and the methane module within the ELMv1-ECA over the tundra ecosystem during the cold season. Some other important processes but not discussed this study, including nutrient dynamics, oxygen reaction and diffusion, etc., are not illustrated here. Grey colours indicate processes that are not actively involved during the cold season over the tundra ecosystem.**
**Orange arrows represent process interactions. Black arrows represent fluxes. Ellipses with thicker red boundaries indicate the modules we modified. Specifically, we revised the new soil water phase-change scheme within the water and heat transfer module, modified carbon decomposition environmental scalar scheme, and incorporated the modified CH4 transport mechanism for the cold-season regime.**





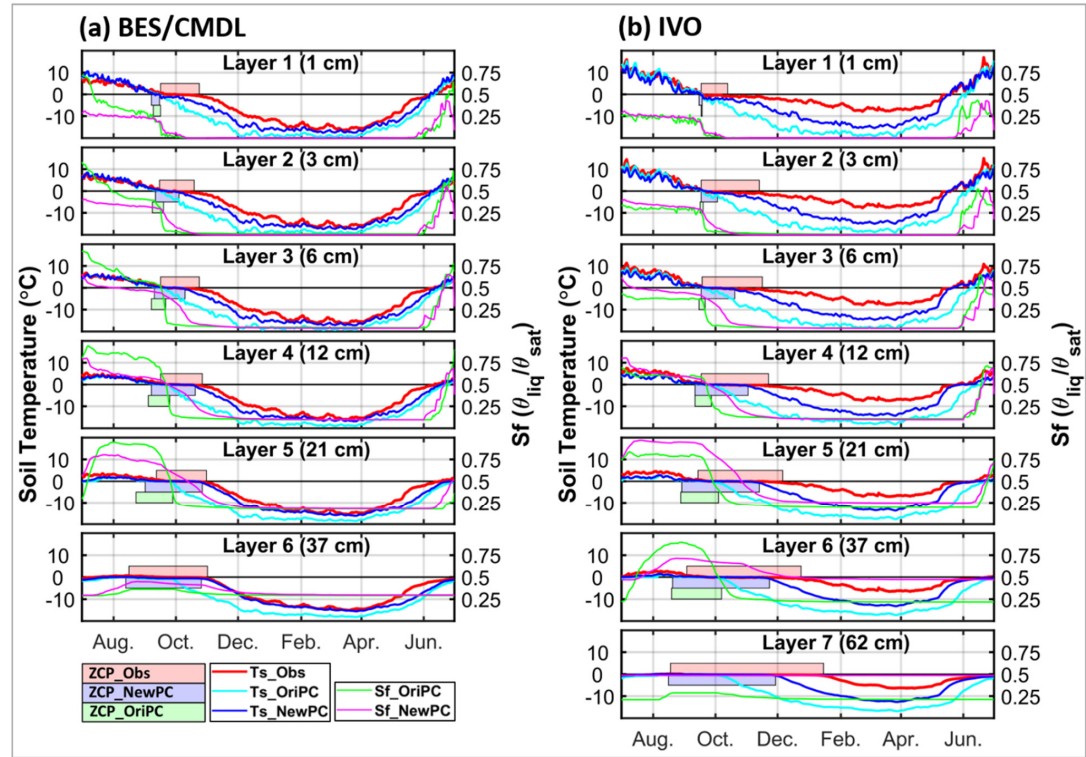


**Figure 3: Comparison of multi-year (2013 - 2017) averaged daily soil temperatures observed (Ts_Obs, red) and simulated with the original (Ts_OriPC, cyan) and improved (Ts_NewPC, blue) phase-change schemes at BES/CMDL (a) and IVO (b). Simulated moisture saturation with the original (Sf_OriPC; green) and improved (Sf_NewPC; magenta) schemes are shown on the right hand axes. The horizontal axes indicates days from July to June, with ticks represent the first day of each month. Hatched areas**
**represent durations of zero-curtain periods observed (ZCP_Obs, orange) and simulated (ZCP_OriPC, green; ZCP_NewPC, blue). No baseline ZCP is shown in the 6th layer for BES/CMDL and the 7th layer for IVO because the maximum annual temperature is below 0°C.**





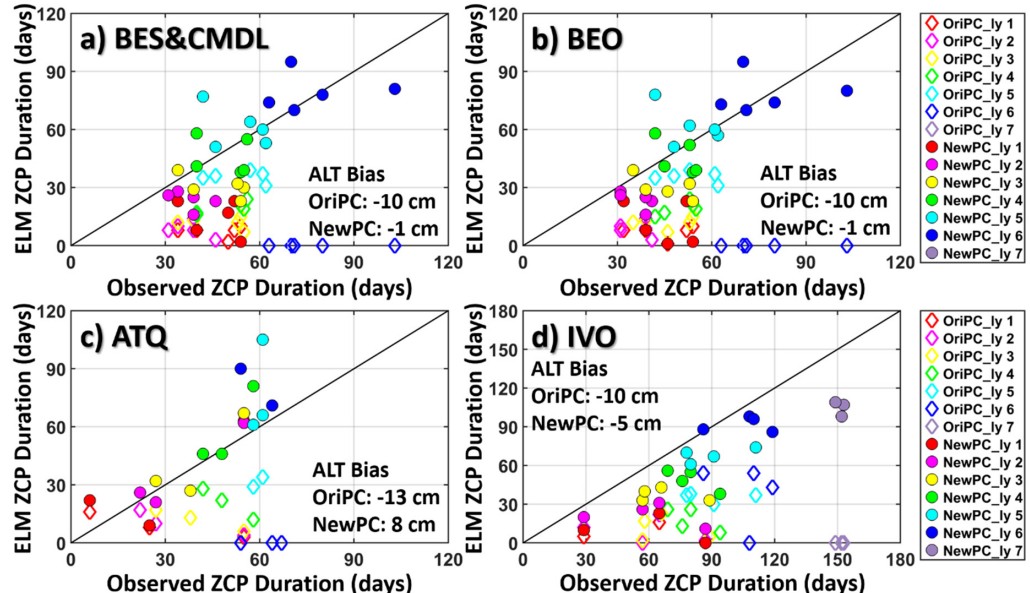


**Figure 4: Comparison between observed and ELMv1-ECA simulated durations of ZCP for the original (OriPC; open diamonds) and improved (NewPC; solid circles) phase-change schemes over four annual cycles (July to June) from 2013 to 2017. "ly" means model layer. Simulated ZCP durations with NewPC demonstrate significant improvements compared to OriPC (solid dots vs. open diamonds), especially for the 4th to the deepest layer above permafrost. Note, a zero days ZCP means that the maximum daily**
**temperature during an annual cycle is below 0°C. The pairs of zero vs. non-zero days ZCP (e.g., OriPC_ly 7 at IVO and OriPC_ly 6 at other sites) indicate that baseline results underestimated ALT. The bias (simulation - observation) of multi-year averaged ALT simulated by the two experiments is provided in each panel.**


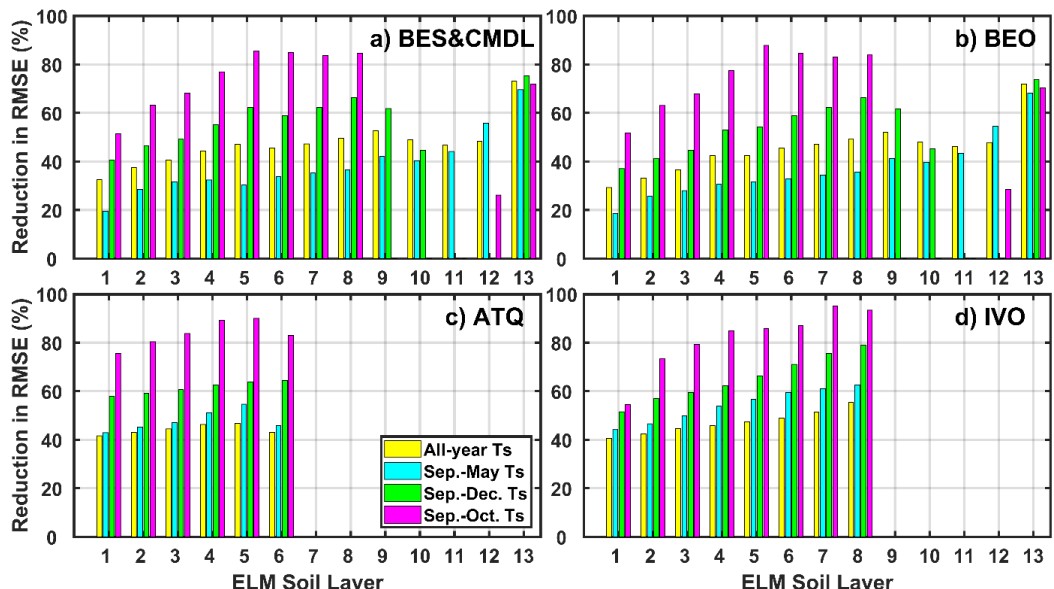

**Figure 5:** Relative improvement in the RMSE of simulated soil temperature with the new phase-change scheme (RMSE_Ts_NewPC) compared to that with the original scheme (RMSE_Ts_OriPC), calculated as 100% × (RMSE_Ts_OriPC − RMSE_Ts_NewPC) / RMSE_Ts_OriPC.





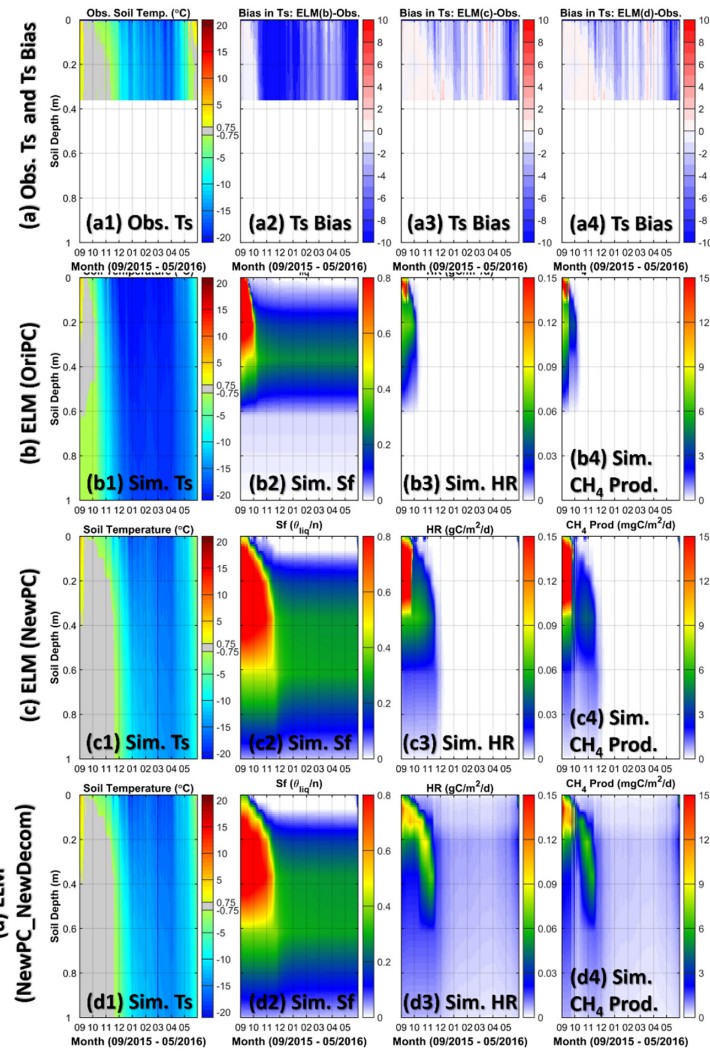

**Figure 6: (a1)** Observed temporal evolution of vertical profiles of soil temperature at ATQ over the cold season from Sep. 1, 2015 to May 31, 2016, and the biases of soil temperatures from three simulations (a2, a3, a4)). ELMv1-ECA simulated baseline evolution of soil temperature (b1), soil liquid water content (b2), heterotrophic respiration, and CH₄ production. (c) Same as (b) with the new phase-change scheme (i.e., NewPC_OriDecom_OriCH4). (d) Same as (c), but using the revised ELMV1-ECA soil moisture-dependency function scheme-2 with built-in oxygen stress (see Figure S1), i.e., NewPC_NewDecom_OriCH4. Soil temperatures within the range of [-0.75°, 0.75°] are coloured by grey, indicating a two-dimensional "zero-curtain zone".

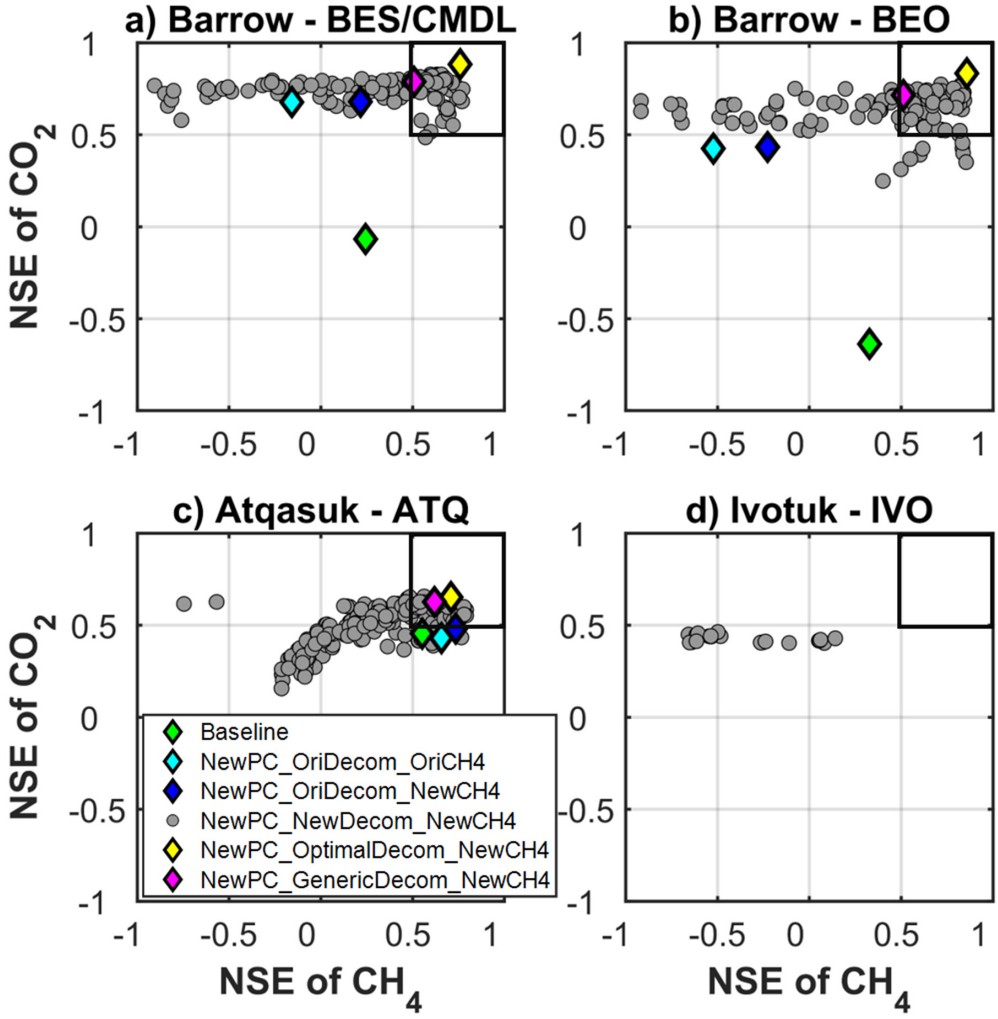

**Figure 7: Scatter plot between the Nash–Sutcliffe Efficiency (NSE) of simulated monthly CH₄ and CO₂ emissions. An ideal simulation has both NSEs of CH₄ and CO₂ as one (i.e., the upper right corner). The boxes encompass simulations with satisfactory performance (NSE > 0.5). Optimal (yellow) – the best simulation for each site; Generic (magenta) – the simulation with a common decomposition scheme that provides best overall performance for all the sites. See Table 2 for the configuration for each experiment. Symbols outside the plotting ranges indicate poor performance, e.g., (-34.9, -0.3) for baseline at IVO, thus are be shown in the figure.**




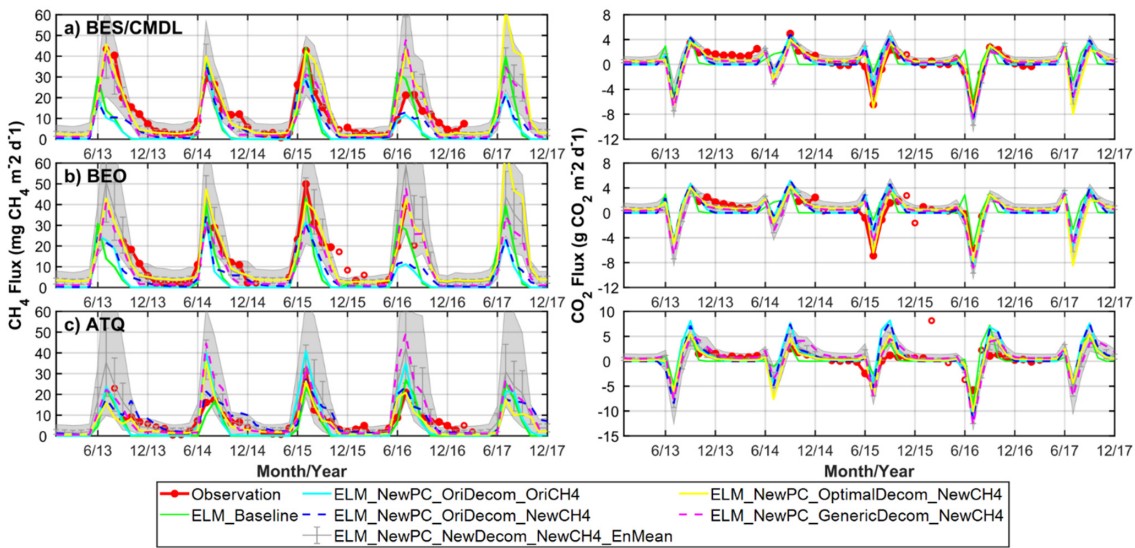

**Figure 8: Observed and simulated monthly CH$_4$ (left) and CO$_2$ (right) net flux with the baseline model (ELM_Baseline) and the experiments with updated models (See Table 2 for the configuration for each experiment). Gray line represents the ensemble mean of simulations within the good performance zone (as shown in Figure 7) with error bars as the standard deviation and the shaded area indicating the minimum-to-maximum bound. Red open circles are observed monthly averages with the number of daily observations less than 10 days, which are not used for the computation in Figure 7. Optimal – the best simulation for each site; Generic – the simulation with a common decomposition scheme that provides best overall performance for all the sites and will be applied to regional simulation.**



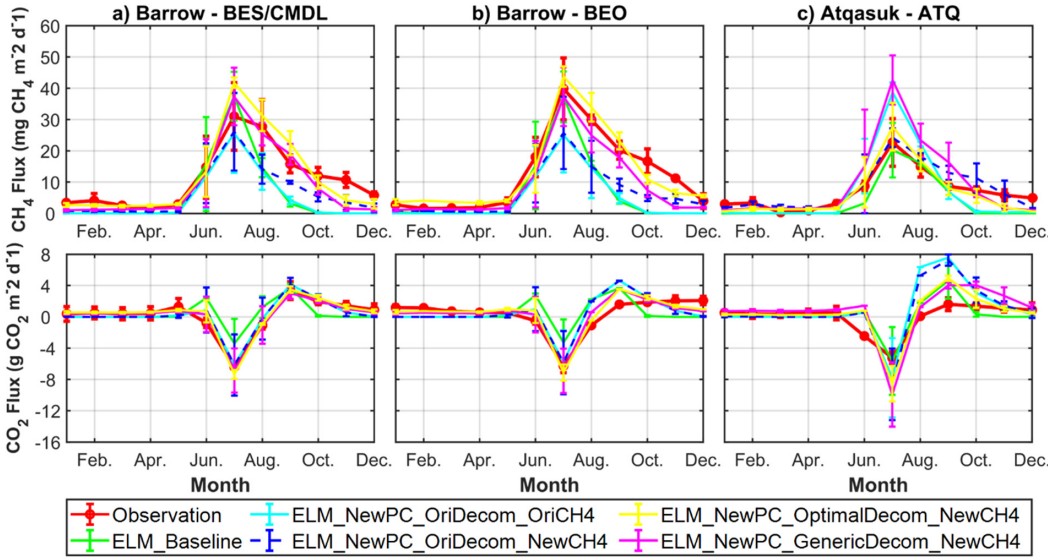


**Figure 9: Comparison of multi-year (2013-2017) averaged monthly mean CH₄ (top) and CO₄ (bottom) net flux from simulations and measurements at BES&CMDL, BEO, and ATQ. The error bars represent standard deviation of monthly mean.**





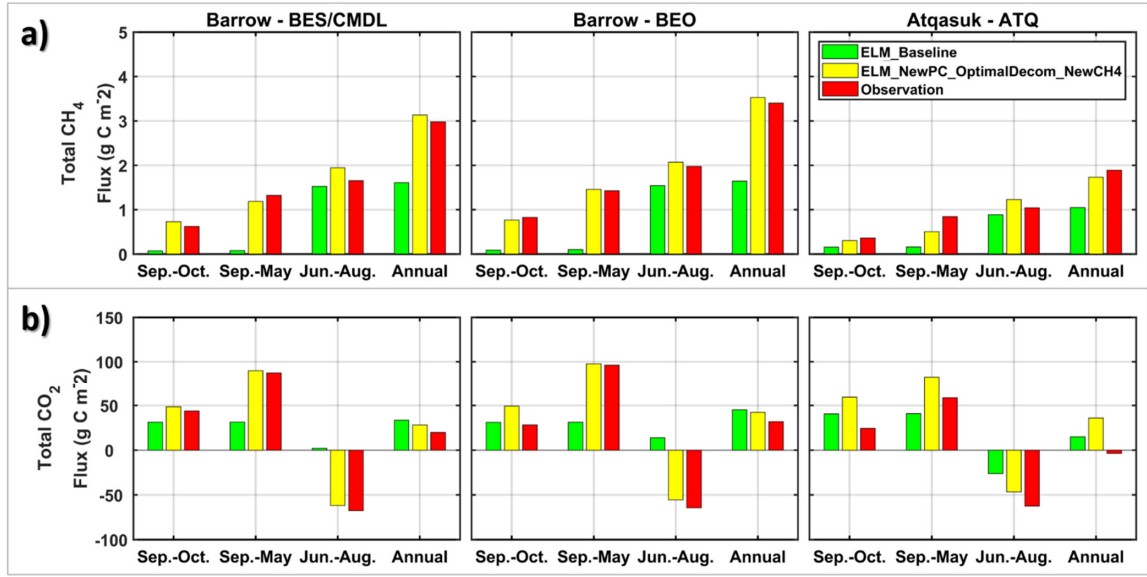

Figure 10: Multi-year (2013-2017) averaged total CH₄ emissions (upper) and CO₂ net fluxes (bottom) during the early cold season (Sep. and Oct.), cold-season period (Sep. to May), warm-season period (Jun. to Aug.), and the annual cycle (Sep. to Aug.) at three of our study sites. Due to the large discontinuity in CO₂ observations, especially over the warm season (shown in Figure 8), the observed annual CO₂ budget is highly uncertain. Still, the cold-season contributions of both CH₄ and CO₂ emissions are greatly improved by the updated ELMV1-ECA (i.e., ELM_NewPC_OptimalDecom_NewCH4).

