# Peer review of "Improved ELMv1-ECA Simulations of Zero-Curtain Periods and Cold-season CH4 and CO2 Emissions at Alaskan Arctic Tundra Sites"

_The Cryosphere, 2020_

## Referee Comment (RC1) · Anonymous Referee #1 · 15 Dec 2020

This study considers cold-season CO2 and methane emissions from Arctic tundra, which has been recently identified as an important component of the tundra carbon budget. In this work, the authors modify an ESM land surface scheme (ELMv1-ECA) to better represent freeze/thaw processes in the soil, to better represent the impact of soil moisture and temperature on soil carbon decomposition, and to allow transport of methane out of the soil during the cold season. This addresses three factors that they identified as causing the model to produce too little carbon emission during the cold season. Namely, that winter soil temperature (and therefore also liquid water) were underestimated, that the CO2 production was too little even with the correct soil temperature, and that methane transport out of the land surface was not possible dur-

ing winter. They find that the model is significantly improved with these modifications, although emissions during the ZCP are now slightly overestimated instead of underestimated. They also look at long term trends.

In general the paper is clearly written and logically ordered. It is clear that the model does improve between the initial and final model versions, and now can simulate cold-season emissions better than previously. I liked the fact that they tested a large number of different functions for the decomposition response, since this is certainly a major source of uncertainty in modelling.

However, before considering this for publication I would ask for some substantial justification and clarity about the changes that have been made. I would also ask for some additions to the text to consider other (potentially) important factors.

General comments:

==============

1) Carbon/substrate. There is plenty of evidence that availability of carbon substrate is important for controlling methanogenesis (e.g. Strom et al 2012), and soil respiration in general (e.g. Brooks et al 2004). This is not discussed in this paper. It is not clear whether the model simulates the soil carbon dynamically, or whether (dynamic or not) the soil carbon takes appropriate values in the model. For one site (IVO) there is some discussion of this (Section 3.2), but it sounds like this soil carbon data is only used to set the soil thermal/hydraulic properties. Does it also form the substrate for soil respiration?

If the substrate is not correctly simulated, then you may compensate for this with incorrect choice of decomposition functions. I would ask for some more analysis of the soil carbon - for example compare it to observed values at the site and identify if this can be a source of bias (I would suggest add to a plot or table in the supplementary material and discuss in the main text).

[Figure]

2) Snow. The main problem with ESM's underestimating winter soil temperature is often related to representation of snow, so I was surprised that this was not discussed in more detail, and only the phase change was considered as leading to underestimated winter soil temperatures (although snow is mentioned once in the results). For example, Burke et al (2020) show the offset between air and soil temperature in CMIP5 and CMIP6 ESM's as a function of snow depth. In models that poorly represent snow, the offset can be up to 10 degree C biased - meaning at 10 degree C cold bias in winter soil temperature. In models that improved their snow scheme between CMIP5 and CMIP6, there is a huge improvement in this. Even in a model that does not represent latent heat *at all*, the winter soil temperature offset against air temperature is substantially smaller than in models with a poor snow insulation scheme. I would therefore strongly recommend that snow is considered in terms of the simulation of winter soil temperatures. I suggest that at the point where snow depth is discussed in the results (see specific comments, below), an assessment of how well the snow is simulated should be presented with supplementary figures.

3) Justification of the phase change modification. While I have no problem that the main modification to the phase change calculation (allowing temperature to fall below zero during the phase change) is physically sound, I am not so convinced by the phase change "efficiency" parameter that was introduced. This is referenced to some papers where such a parameter was included in a model previously, but those papers are extremely brief in the justification of this and there is no reference to some observation or physical theory. Additional justification is therefore required for this "efficiency" parameter (or removal of this parameter if it is not fully justified).

4) Environmental modifiers. The soil moisture function is modified in two ways: 1 to decrease respiration at high water contents, which replaces the oxygen-availability modifier, and 2 to continue to have respiration at zero water contents. Firstly, why would you replace the process-based oxygen availability with an empirical function that would presumably represent the process less well? And secondly, setting non-

zero respiration at zero water contents is dubious and the functions chosen (shown in Fig S1) are strange-looking. There is no evidence that respiration occurs at zero water contents, rather that it continues over winter because there is non-zero liquid water in frozen soil. It would make much more sense to change the function so that it reaches zero at zero water contents, i.e. shift the curves to the left. This would also look a lot more like existing literature, e.g. Yan et al 2018. So, I am not convinced by these modifications.

5) CH4 cold season transport. Again this is poorly justified and the equations are missing. In the appendix line 700-702 it states that "We integrate the emissions from ice cracks and remnants of aerenchyma tissues with (Eq. C14) by removing temperature limitation and applying a small Taere during winter time" Firstly, there was no mention of temperature limitation, so what does this part refer to? Secondly, what is "a small Taere"? (ie what is the value and why did you choose it?). It would really be useful to give the equation that you use in the model, instead of just this unclear description. In the methods it is justified by reference to a paper that there could be more conduction of methane through snow. However, the change made to the diffusion inside the soil is set as an arbitrary value and a sufficient justification is not given for changing it. Potentially, increasing Taere would increase the emissions enough (if you choose the right value) that this non-justified change to diffusion rate would not be needed.

6) Missing out of IVO. It is shown that IVO cannot simulate reasonable methane emissions even with the correct temperature and moisture, however CO2 emissions did not suffer from this problem. Therefore I don't fully understand why CO2 from IVO cannot be included in the analysis. The optimisation would have to be done only on CO2 but it could still be optimised, is that right?

7) Additional analysis: For further investigation of the temperature/moisture functions in frozen conditions, you could plot the emissions against temperature and moisture, instead of over time. Then you can see if the models and observations are producing similar functions. (This would show up, for example, if there is hysteresis in the observations which would make it difficult for any single function to capture the dynamics, and would be worthwhile to know.)

Specific comments

=============

Abstract

Line 16-17 "simulated cold-season emissions at three tundra sites were improved by 84% and 81%" - it is not clear what metric the 84% and 81% refer to, is this the mean absolute error? Please specify.

Line 17-19 "...zero-curtain period in Arctic tundra, accounted for more than 50% of the total emissions" This statement is slightly misleading. This is the case in the model, but the study showed that this part was overestimated compared to the observations. I would therefore add something like "in the model, compared with around 45% (30-60%) in the observations"

Introduction

Lines 60-62 " However, current land models tend to significantly underestimate soil temperature during the cold season over permafrost regions (Dankers et al., 2011; Tao et al., 2017; Nicolsky et al., 2007; Yang et al., 2018b). One possible reason is that many land models fail to appropriately account for the latent heat released during soil water freezing" It is true that many land surface models did underestimate soil temperatures but, more recently, improved snow schemes have removed a lot of this problem. For example, your first reference Dankers et al (2011) has a followup study Burke et al (2013) which includes a multilayered snow scheme and removes the majority of the winter cold bias - although a small cold bias remains. I highly recommend adding some discussion of snow here to make it clear that the latent heat is not the only (or even the biggest) factor. Most recent LSM's (e.g. in CMIP6) do represent latent heat, if not particularly well, I suggest clarifying that to "One possible reason is that while

many land models account for latent heat released during soil water freezing, they do not treat and distribute this heat appropriately"

Lines 69 "many land models cannot accurately capture the ZCP length due to their underestimation of soil temperatures" This is not really an accurate statement. Many land models cannot accurately capture the ZCP length (true), but this is because they don't have enough soil moisture or an adequate representation of latent heat, not "due" to underestimated temperatures. Rather, underestimated temperatures can arise as a *result* of not simulating the ZCP.

Study sites and data

Line 93. " CARVE $CO_2$ measurements were not available;" should this be "...were not available from 2015-2017;" ? Currently this part is unclear. Line 116-117 "Due to the discontinuity of observed soil moisture over time and along with the vertical depth, evaluating ELMv1-ECA simulated liquid water content at layer node-depth was limited." This sentence does not make sense to me, please clarify.

Methodology

Line 152-153 "The underlying assumption here is that the liquid water of soil resists freezing as the freezing process proceeds and Sfnliq,i decreases, analogous to how dry soils resist getting drier due to capillary force." This is the explanation given for the efficiency factor (see comment 3, above): However, this capillary force in freezing soils is represented by the non-zero liquid water contents at sub-freezing temperatures, and it is not clear to me that it needs an additional factor. The efficiency factor, I guess (although it is not clear what it actually does - see next comment) corresponds to a 'loss' of some of the energy produced by latent heat. It does not make sense that energy would just disappear. Please explain/justify.

Line 153-154 "We applied the phase change efficiency to the initially estimated energy and mass change involved, i.e., ðİŘżàŕIJ and ðİŘżàŕă (see (Eq. A4) in the Appendix)".

It is not clear what 'applied' means here, did you multiply some part of these equations by the efficiency factor? The easiest thing to do would be to include the equations in the Appendix that you actually used (i.e. rewrite those equations with the efficiency factor included, instead of leaving it to our imagination).

Table S2: I am missing where the moisture functions ModifiedELM‐S1 ModifiedELM‐S2, etc are documented?

Line 264-266 "We confirmed that ELMv1-ECA's PFT dataset was a good compromise between representing the site-scale ecosystem and other global parameters and surface datasets within ELM. " Firstly, what is ELMv1-ECA's PFT dataset? This is not mentioned. Secondly, how did you assess whether it was 'good'? I recommend adding more information here.

Line 266-267 "The simulated saturated and unsaturated CH4 emissions were weighted with the estimated inundation fractions at the footprint of ABoVE eddy-covariance flux towers" Surely the *unsaturated CH4 emissions should be weighted with the *non-inundated fraction in the footprint? I guess this is probably what you did, it's just not written very clearly, it currently sounds like both saturated and unsaturated CH4 emissions were multiplied by the inundated fraction.

Readers will not know that the model simulates methane separately from saturated and unsaturated grid cell fractions, therefore I suggest making that point here.

Results

Line 326-343 Here you talk about the improvement to the ZCP. Looking at the plots, there is a great improvement in deeper soil layers but not so much in the surface (for 3/4 sites). I suggest that the text should recognise this fact about the surface being less well simulated.

Line 346 Reference to Figure 3 should be Figure 4.

Line 357-358 "The deeper active layer simulated by NewPC implies more soil water

storage capacity, resulting in lower soil moisture in shallow soil layers and higher soil water in deep layers" This pattern is not really seen with most of the sites, either the new simulation seems to have lower soil moisture in general, or in the case of IVO it is greater or similar in almost every layer in the new simulation. There is also the claim of soil moisture being improved - this is true because the timing of thaw and freeze-up is better, but actually the level of saturation in general seems to now be lower and in several cases the old scheme was better in that regard. This is just a suggestion, but I am aware of more than one land surface scheme that has found their scheme of dealing with saturation of soil moisture leads to water being forced out of the top of the soil during the freeze-up period. I was just wondering if simulating a longer ZCP might lead to more water being lost in this way, and would therefore explain why the new model is drier. There are several possibilities, of course!

Line 383-384. As I discussed above, the snow is important and I suggest that this is the place to present some additional analysis rather than simply referring to "underestimated snow depth (not shown)".

Line 416, having checked that using observed soil moisture and temperature does not improve the CH4 simulation, saying that including advective heat transport would likely improve the simulations is surely incorrect, since this would just improve the soil temperature, which you found did not help. I would also be surprised if a better wetland simulation would help if using observed soil moisture did not improve the simulation. Geological seepage is certainly a possibility though.

Line 420-423. This is missing the information that the performance of CH4 is degraded at the BES/CMDL and BEO sites. It's somewhat misleading to only mention the improvements.

Line 442-444. This part is unclear. When you say "soil properties", do you mean soil thermal/hydraulic properties? And is the improvement of the ELMv1-ECA's moisture scalars due to the function being based on the suction rather than the volumetric soil

moisture content? Please clarify this. Also, please give evidence that ELMv1-ECA "reasonably explained the varying influence along the vertical soil profile".

Line 444 "Thus, the simulations..." I suggest removing 'thus' because overestimation isn't implied from the previous sentence.

Line 450 "assigns small thresholds for the moisture scalar" this is unclear. Did you mean "assigns small minimum values for the moisture scalar"? Line 451: Same problem as 450.

Line 457-459 ", at ATQ, where cold-season temperatures are relatively warmer than at BES/CMDL and BEO, simulations with the original ELMv1-ECA environmental modifier (i.e., "NewPC_OriDecom_NewCH4"; discussed in Section 3.1.2), already released much more CO2 and CH4 throughout the cold season than in the baseline simulations, " Can you add a reference to a Table or Figure that shows this happens more at ATQ than the other sites? It isn't very clear to me on Figure 8. And in fact on Figure 6 (c3 and c4), it looks like the cold season production of CO2 and CH4 still goes to zero at ATQ with NewPC_OriDecom_NewCH4.

Line 471. vary -> varies

Line 471-474 " For cold sites (i.e., BES/CMDL and BEO), the sensitivity of simulated CH4 to Q10 values is larger than the sensitivity of CO2 net flux to Q10 because cold temperature suppresses vegetation growth (i.e., CO2 uptake); while for the warm site (i.e., ATQ), both CH4 and CO2 net flux are very sensitive to the Q10 values." Can you refer here to some numbers/figures that show this? I see that in Table S4 this is apparent if you compare the lines with the same soil moisture function but different Q10's.

Line 485/Section 4.3. I suggest you start this section with a clarification that throughout this section you are analysing the results with optimal decomposition scheme for each site (and therefore different parameters are used for each site).

Line 496 slightly -> slight

Line 511-512 "We find that the simulated cold-season CO2 emissions were larger than the warm-season CO2 net uptake at all three sites" Please specify during which time period. (Presumably they are in balance during the spinup, but will become out of balance later in the simulation due to changing climate, so it makes sense to note the time period here)

Summary

I suggest that you additionally mention the potential issues of using the heterotrophic respiration to estimate CH4 production. For example, this means that CH4 emissions may drop as the soil becomes more saturated (once soil moisture passes the optimum), whereas in fact the highest CH4 emissions should be in saturated conditions.

Line 529 "by updating upper boundary resistance" Was this the only change? What about the change you made to the diffusion through the soil? I don't think that is related to the upper boundary? Please check this to make sure it's summarizing accurately.

Line 546 "the identified an" -> "the identified"

Line 562 add "due to microbial dynamics" or similar, for clarity

Line 571-573 "The increasing rate of cold-season heterotrophic respiration (releasing CO2) may become larger than the trend of warm-season vegetation CO2 uptake under future climate" In fact in your simulations, the cold season respiration already became larger than the warm season CO2 uptake by 2017, is that right? This point could be made stronger with that information.

Appendix Eq. A3. What does * mean in this equation? Eq. A7. as already discussed in my comments on the Methodology, you need to show where/how these factors are applied in the model - via equations would be easiest. Line 642. This equation isn't entirely consistent with equation A3, the $10^3$ is on the bottom and g is missing. Can you check both of these? Line 692. It is not entirely clear what A(z) represents, is this the

total methane emission to the atmosphere or just the part from aerenchyma? Line 700: Apologies if this is common knowledge but I don't know what "amount of carbon per tiller" means. Is this correct? Line 700-702: This needs more explanation/equations, see comment (5) in general comments, above. Line 710: "Table 2" should be Table 1, I think Line 710: Please specify which parameter in the equations you are changing. The table refers to it as "scale_factor_gasdiff_snow" and it's not clear where this fits in Eq. C13 (if it all) Line 711: presents -> is present.

Figures

General: Firstly, it is common to plot the observations in black and the model versions in colours (or at least use a different style of line), which I would recommend here since it would add clarity to the plots. Secondly, there appears to be a slight difference in the $CO_2$ when the methane transport modifications are introduced, particularly for ATQ. I did not see any way that the methane transport would influence the $CO_2$ simulation - could you explain this difference?

Figure 1 is not super clear which labels are refering to which sites, since there are more labels than red dots. Could you add lines or arrows to indicate for certain which site is in which location.

Figure 2 is a bit of a mess and it does not seem to be logically organised. For example, the split between green circles appears to be between methane and "every other form of carbon", perhaps it would make sense to separate vegetation and soil (non root) carbon? Most of the arrows are brown and seem to represent "some kind of influence". To me it is important to show the flows of carbon between the different spheres, some of which is shown in black (e.g. $CO_2$ emission), some in brown (e.g. heterotrophic respiration producing $CH_4$), and some not shown at all, such as the flow of carbon between plants and soil. This diagram needs to be revisiting to get a complete and coherent presentation.

Papers mentioned

Burke et al 2013 https://link.springer.com/article/10.1007%252Fs00382-012-1648-x

Burke et al 2020 https://tc.copernicus.org/articles/14/3155/2020/

Brooks et al 2004 https://onlinelibrary.wiley.com/doi/full/10.1111/j.1365-2486.2004.00877.x

Strom et al 2012 https://www.sciencedirect.com/science/article/abs/pii/S0038071711003385

Yan et al 2018 https://www.nature.com/articles/s41467-018-04971-6

---

## Referee Comment (RC2) · Anonymous Referee #2 · 21 Dec 2020

The manuscript by Tao et al. describes an improved capacity of the ELM land surface model to simulate the zero-curtain period and cold season greenhouse gas emissions. The paper is well-written and the changes made to the model are well-described. I don't see large shortcomings to this paper but, like the other reviewer, it would be nice to have a few more clarifications on why certain approaches were chosen and to place the results in a broader context.

First of all, the model is only tested on four sites in Alaska. Two are from the same area, while the other two are further inland. I'm not convinced that this climatic gradient is sufficient to capture the dynamics of the cold season across the Arctic, which is the

stated goal by the authors for their next paper. Especially since the model does not capture the soil temperature during the cold season at IVO. This may be due to the model setup (e.g. soil conditions or atmospheric forcing), but could also be due to an incorrect simulation of the insulation of the snow as suggested by the other reviewer. In any case, this does not add confidence that the model will perform well in, for example, central Siberia or in the sub-Arctic, where winter conditions are quite different from the north slope of Alaska. This regional bias needs to be considered in the text since it is essential to judge the performance of the model.

Second, the simulation of cold season greenhouse gas emissions is much improved but, again, with only a few sites used for validation this may be getting the right numbers for the wrong reasons, when the model has been specifically optimized for these sites. The addition of cracks and plant remnants to act as conduits to the atmosphere makes sense, but this is a rudimentary solution that does not enable the simulation of sudden bursts of $CO_2$ and $CH_4$ which have been observed across the Arctic during the cold season – including at Barrow (Mastepanov et al., 2008; Pirk et al., 2017; Raz-Yaseef et al., 2017). A discussion on why the model is not able to do this, and how this may lead to a systematic bias would be warranted.

Finally, the paper is incredibly detailed, which is generally welcome, but in this case there are simply too many figures and tables. The information presented in Figure 5 overlaps with Figure 3 and Table 4, for example. I suggest that some of these figures and tables are moved to the supplemental information, especially when they're only briefly discussed in the text.

Also, some of the figures are incredibly busy because several parameters are plotted together but this makes it confusing to me what I'm looking at without continuously checking the legend. The colors are hard to distinguish from each other, especially the yellow color when printed. It would also help if the observations are plotted with a clear black line or dashed vs continuous, for example, and that soil moisture and temperature are also plotted with different line types.
Minor comments:

Page 4, Line 110: were these gaps large? If gaps were only a few days this is fine, but it would be good to know if weeks or months of data needed to be gap-filled.

Page 6, line 176-177: no need to specify that the 'S' stands for supplemental. This is rather standard knowledge.

Page 13, line 405: it's unclear to me why there's an ensemble of grey dots for the NewPC_NewDecom_NewCH4 but not for the other dots? This is not well-described in the caption or the text.

Page 18, line 562: please elaborate on why the single static multiplicative function would not be appropriate.

References

Mastepanov, M., Sigsgaard, C., Dlugokencky, E. J., Houweling, S., Ström, L., Tamstorf, M. P. and Christensen, T. R.: Large tundra methane burst during onset of freezing, Nature, 456(7222), 628–630, doi:10.1038/nature07464, 2008. Pirk, N., Mastepanov, M., López-Blanco, E., Christensen, L. H., Christiansen, H. H., Hansen, B. U., Lund, M., Parmentier, F.-J. W., Skov, K. and Christensen, T. R.: Toward a statistical description of methane emissions from arctic wetlands, Ambio, 46(1), 70–80, doi:10.1007/s13280-016-0893-3, 2017. Raz-Yaseef, N., Torn, M. S., Wu, Y., Billesbach, D. P., Liljedahl, A. K., Kneafsey, T. J., Romanovsky, V. E., Cook, D. R. and Wullschleger, S. D.: Large CO2 and CH4 emissions from polygonal tundra during spring thaw in northern Alaska, Geophysical Research Letters, 44(1), 504–513, doi:10.1002/2016GL071220, 2017.

---

## Author Comment (AC1) · 1 Feb 2021

The comment was uploaded in the form of a supplement:
https://tc.copernicus.org/preprints/tc-2020-262/tc-2020-262-AC1-supplement.pdf

---

## Author Comment (AC2) · 1 Feb 2021

**Response to the Reviewer #2:**

We thank the reviewer for the constructive comments and suggestions that have helped us rethink and improve the manuscript. We will revise the manuscript according to the reviewer's comments (see point-by-point responses below). Throughout this document, the reviewer's comments are reproduced in their entirety in black, and our responses are given directly afterward in blue.

The manuscript by Tao et al. describes an improved capacity of the ELM land surface model to simulate the zero-curtain period and cold season greenhouse gas emissions. The paper is well-written and the changes made to the model are well-described. I don't see large shortcomings to this paper but, like the other reviewer, it would be nice to have a few more clarifications on why certain approaches were chosen and to place the results in a broader context.

We appreciate the reviewer's constructive comments. We will provide particularly detailed elaboration for the issues pointed out by the reviewer in the revised manuscript as discussed by the following responses.

First of all, the model is only tested on four sites in Alaska. Two are from the same area, while the other two are further inland. I'm not convinced that this climatic gradient is sufficient to capture the dynamics of the cold season across the Arctic, which is the stated goal by the authors for their next paper. Especially since the model does not capture the soil temperature during the cold season at IVO. This may be due to the model setup (e.g. soil conditions or atmospheric forcing), but could also be due to an incorrect simulation of the insulation of the snow as suggested by the other reviewer. In any case, this does not add confidence that the model will perform well in, for example, central Siberia or in the sub-Arctic, where winter conditions are quite different from the north slope of Alaska. This regional bias needs to be considered in the text since it is essential to judge the performance of the model.

**R2C1:** We agree with the reviewer that the site number is limited. We would also like to test more sites, however, sites over permafrost regions that have all the necessary measurements needed for this study, including snow depth, soil temperature, soil moisture, year-round $CO_2$ and $CH_4$ fluxes, are quite rare. We did test the updated ELM at Alaska SNOTEL sites and all the cold-season $CO_2$ flux sites over pan-Arctic permafrost regions as reported by Natali et al. (2019). But, we only found one pair of SNOTEL sites (AK-968) and cold-season $CO_2$ flux sites that are co-located with each other and have reasonable observations during an overlapping period. We will definitely explore more sites in the future.

At IVO, the model well reproduced cold season soil temperatures, as shown in Figures 3, 4, and 5. We agree with the reviewer about the importance of soil conditions, atmospheric forcing, and snow conditions. We tested multiple reanalysis forcing, including CRUNCEP (Climatic Research Unit and NCEP Reanalysis; QianFilled) and GSWP3 (Global Soil Wetness Project Phase 3), and we will provide simulation results at the tested tundra sites with other climate forcing in the supplementary file. We will also provide a comparison of simulated snow depths against measurements (which show problematic measurements, though) and sensitivity analysis of simulated carbon fluxes to snow conditions in the supplementary file. Please also see our response to Reviewer#1 (**R1C2**).

In a recent paper focusing on regional simulation over Alaskan Arctic tundra, we used the generic decomposition scheme identified in Figure 7 that provided the best overall performance for all the sites. We found that the updated ELMv1a demonstrates significantly improved performance in the

simulated regional mean of cold-season $CO_2$ emissions over the Alaska North Slope tundra, showing a 55% reduction in RMSE compared to the bassline results (0.14 versus 0.31 gC m$^{-2}$ day$^{-1}$) (Tao et al., 2020).

We also tested the updated ELM (again with the generic decomposition scheme) with multiple reanalysis forcing datasets over pan-Arctic permafrost regions. We found that, compared to a machine-learning derived spatial dataset of $CO_2$ emissions based on ground in situ measurements (denoted N2019 here) (Natali et al., 2019; Watts et al., 2019), the simulated $CO_2$ emissions results driven by GSWP3 show generally smaller biases over pan-Arctic permafrost than that driven by CRUJRA (**Figure R2.1**). We will discuss these biases and possible reasons in our following paper (Tao et al., 2021).

[Figure]

**Figure R2.1:** Bias in multi-year averaged (2003 to 2017) spring $CO_2$ Emissions over pan-Arctic permafrost. Left: Bias = Simulations by ELMv1 driven by CRUJRA – N2019; Right: Bias = Simulations by ELMv1 driven by GSWP3 – N2019. (Preliminary results subject to change.)

We will also add into the revised manuscript some discussion about the representativeness of the tested tundra sites and how the generic decomposition scheme identified here can be transferred to a larger scale across a variety of climate and ecosystem gradients.

Second, the simulation of cold season greenhouse gas emissions is much improved but, again, with only a few sites used for validation this may be getting the right numbers for the wrong reasons, when the model has been specifically optimized for these sites. The addition of cracks and plant remnants to act as conduits to the atmosphere makes sense, but this is a rudimentary solution that does not enable the simulation of sudden bursts of CO2 and CH4 which have been observed across the Arctic during the cold season – including at Barrow (Mastepanov et al., 2008; Pirk et al., 2017; Raz-Yaseef et al., 2017). A discussion on why the model is not able to do this, and how this may lead to a systematic bias would be warranted.

**R2C2:** We were aware of the sudden bursts of $CO_2$ and $CH_4$ during the freeze-up period because the gases are pushed out of freezing soils. Currently, we mimicked this mechanism by preventing $CO_2$ and $CH_4$ from dissolving in the soil ice fraction (Riley et al., 2011). As suggested by the reviewer, we will add discussion on how we simulate this mechanism and why it currently cannot well capture the sudden burst, and also our plan to address this issue in the future.

Finally, the paper is incredibly detailed, which is generally welcome, but in this case there are simply too many figures and tables. The information presented in Figure 5 overlaps with Figure 3 and Table 4, for example. I suggest that some of these figures and tables are moved to the supplemental information, especially when they're only briefly discussed in the text.

**R2C3:** We agree with the reviewer about the repeated information from Figure 5 and Table 4, and thus we will move Table 4 to the supplementary file and retain Figure 5 in the manuscript. We use Figure 3 to analyze the simulated freezing process and how the revised soil water phase-change scheme improves soil temperature simulations, explicitly highlighting the better simulated zero-curtain periods, which is critical to our manuscript. We will also remove Figure 2 (see our response to Reviewer #1 R1C49).

Also, some of the figures are incredibly busy because several parameters are plotted together but this makes it confusing to me what I'm looking at without continuously checking the legend. The colors are hard to distinguish from each other, especially the yellow color when printed. It would also help if the observations are plotted with a clear black line or dashed vs continuous, for example, and that soil moisture and temperature are also plotted with different line types.

**R2C4:** Thank you for the suggestions. We will modify our figures accordingly. Specifically, we will replot time-series plots (including Figures 3, 8, and 9), using black lines for observations, and also change yellow lines to other colors or in different line styles. Particularly for Figure 3, we will change the soil moisture lines to another line type with contrasting colors.

Minor comments:
Page 4, Line 110: were these gaps large? If gaps were only a few days this is fine, but it would be good to know if weeks or months of data needed to be gap-filled.

**R2C5:** The original sentence is copied below:

"We first filled missing gaps vertically by fitting a polynomial to the soil temperature profile (Kurylyk and Hayashi, 2016) on a daily scale, then screened out outliers by examining the daily time series."

The ABoVE/CARVE in situ measurements are available at depth 5 cm, 10 cm, 15 cm, 20 cm, 30 cm, and 40 cm, and UAF GIFL soil temperatures are also available at various depths. Here, by "gaps" we meant the discontinuities in measurements along the vertical soil depths. For instance, sometimes the measurements at 20 cm are missing, then we filled this missing data by fitting a polynomial to the soil temperature profile, i.e., measurements available at other depths. We only perform this gap-filling if we have at least one measurement at depths above the missing measurement depth and at least one measurement at depths below the missing depth. For gaps in time, we did not perform gap-filling in case introducing artifacts.

Page 6, line 176-177: no need to specify that the 'S' stands for supplemental. This is rather standard knowledge.

**R2C6:** We will remove this.

Page 13, line 405: it's unclear to me why there's an ensemble of grey dots for the NewPC_NewDecom_NewCH4 but not for the other dots? This is not well-described in the caption or the text.

**R2C7:** The grey dots represent all the tested (200) new decomposition schemes listed in Table S2 as indicated in the annotation of Table 2. We will add clear clarification in the context and the caption of Figure 7. Thanks for the suggestion.

Page 18, line 562: please elaborate on why the single static multiplicative function would not be appropriate.

**R2C8:** We will add more elaboration on this and change the sentence as below.

"In addition, the single static multiplicative function used to parameterize the impact of environmental conditions on respiration might not be appropriate, because the environmental impact also depends on maximum respiration rate, soil texture, soil carbon content, and microbial biomass (Tang and Riley, 2019)."

**References**
Mastepanov, M., Sigsgaard, C., Dlugokencky, E. J., Houweling, S., Ström, L., Tamstorf, M. P. and Christensen, T. R.: Large tundra methane burst during onset of freezing, Nature, 456(7222), 628–630, doi:10.1038/nature07464, 2008.

Pirk, N., Mastepanov, M., López-Blanco, E., Christensen, L. H., Christiansen, H. H., Hansen, B. U., Lund, M., Parmentier, F.-J. W., Skov, K. and Christensen, T. R.: Toward a statistical description of methane emissions from arctic wetlands, Ambio, 46(1), 70–80, doi:10.1007/s13280-016-0893-3, 2017.

Raz-Yaseef, N., Torn, M. S., Wu, Y., Billesbach, D. P., Liljedahl, A. K., Kneafsey, T. J., Romanovsky, V. E., Cook, D. R. and Wullschleger, S. D.: Large CO2 and CH4 emissions from polygonal tundra during spring thaw in northern Alaska, Geophysical Research Letters, 44(1), 504–513, doi:10.1002/2016GL071220, 2017.

**References For Author's Comments**

Kurylyk, B. L., and Hayashi, M.: Improved Stefan Equation Correction Factors to Accommodate Sensible Heat Storage during Soil Freezing or Thawing, Permafrost Periglac, 27, 189-203, 2016.
Riley, W. J., Subin, Z. M., Lawrence, D. M., Swenson, S. C., Torn, M. S., Meng, L., Mahowald, N. M., and Hess, P.: Barriers to predicting changes in global terrestrial methane fluxes: analyses using CLM4Me, a methane biogeochemistry model integrated in CESM, Biogeosciences, 8, 1925-1953, 10.5194/bg-8-1925-2011, 2011.
Tang, J. Y., and Riley, W. J.: A Theory of Effective Microbial Substrate Affinity Parameters in Variably Saturated Soils and an Example Application to Aerobic Soil Heterotrophic Respiration, J Geophys Res-Biogeo, 124, 918-940, 2019.
Tao, J., Zhu, Q., Riley, W. J., and Neumann, R. B.: Warm-season net CO2 uptake outweighs cold-season emissions over Alaskan Arctic tundra under current and RCP8.5 climate Environmental Research Letters, (Under Review), 2020.
Tao, J., Zhu, Q., Riley, W. J., and Neumann, R. B.: Snow-to-Rain Shifts Regulate Cold-Season Carbon Emissions From pan-Arctic Permafrost, TBD. (In Preparation), 2021.
Watts, J. D., Natali, S., Potter, S., and Rogers, B. M.: Gridded Winter Soil CO2 Flux Estimates for pan-Arctic and Boreal Regions, 2003-2100, https://doi.org/10.3334/ORNLDAAC/1683, 2019.

---

## Author Response (AR1)

**Response to the Reviewer #1:**

We thank the reviewer for the constructive comments and suggestions that have helped us rethink and improve the manuscript. We have revised the manuscript according to the reviewer's comments (see point-by-point responses below). Note there are some differences between the responses here and that within the authors' comments uploaded earlier. Specifically, we now decided to follow the reviewer's suggestion on replacing our originally modified ELM moisture scalars with new scalars and then conducted more simulations together with sensitivity analysis to parameters related to carbon decomposition and methane model. We have updated our results and discussions accordingly; our major conclusions about the model improvements in simulating soil temperature and zero-curtain period remain the same, and the improvements regarding simulating  $CO_2$  and  $CH_4$ fluxes still hold, although with different optimized parameterizations.

For reference, our response to comment "n" by the reviewer is labeled "R1C[n]" where R1 represents Reviewer #1. Throughout this document, the reviewer's comments are reproduced in their entirety in black, and our responses are given directly afterward in blue.

This study considers cold-season CO2 and methane emissions from Arctic tundra, which has been recently identified as an important component of the tundra carbon budget. In this work, the authors modify an ESM land surface scheme (ELMv1-ECA) to better represent freeze/thaw processes in the soil, to better represent the impact of soil moisture and temperature on soil carbon decomposition, and to allow transport of methane out of the soil during the cold season. This addresses three factors that they identified as causing the model to produce too little carbon emission during the cold season. Namely, that winter soil temperature (and therefore also liquid water) were underestimated, that the CO2 production was too little even with the correct soil temperature, and that methane transport out of the land surface was not possible during winter. They find that the model is significantly improved with these modifications, although emissions during the ZCP are now slightly overestimated instead of underestimated. They also look at long term trends.

In general the paper is clearly written and logically ordered. It is clear that the model does improve between the initial and final model versions, and now can simulate cold season emissions better than previously. I liked the fact that they tested a large number of different functions for the decomposition response, since this is certainly a major source of uncertainty in modelling.

However, before considering this for publication I would ask for some substantial justification and clarity about the changes that have been made. I would also ask for some additions to the text to consider other (potentially) important factors.

We appreciate the reviewer's constructive comments. We have provided particularly detailed elaboration for the issues pointed out by the reviewer in the revised manuscript as discussed by the following responses.

**General comments:**

1) Carbon/substrate. There is plenty of evidence that availability of carbon substrate is important for controlling methanogenesis (e.g. Strom et al 2012), and soil respiration in general (e.g. Brooks et al 2004). This is not discussed in this paper. It is not clear whether the model simulates the soil carbon dynamically, or whether (dynamic or not) the soil carbon takes appropriate values in the model. For one site (IVO) there is some discussion of this (Section 3.2), but it sounds like this soil

carbon data is only used to set the soil thermal/hydraulic properties. Does it also form the substrate for soil respiration?

If the substrate is not correctly simulated, then you may compensate for this with incorrect choice of decomposition functions. I would ask for some more analysis of the soil carbon - for example compare it to observed values at the site and identify if this can be a source of bias (I would suggest add to a plot or table in the supplementary material and discuss in the main text).

**R1C1:** As we pointed out in the manuscript (lines X - Y), the dependencies of soil thermal and hydraulic properties on soil organic carbon are embedded in the model via linear relationships (Lawrence and Slater, 2008). The relationships between soil properties and organic carbon content are initialized at the beginning of simulations.

We also agree with the reviewer about the importance of carbon substrate on influencing methanogenesis activity and soil respiration, and ELMv1-ECA does account for these impacts. We now have clearly stated that in the revised manuscript (within the Appendices).

"ELMv1-ECA explicitly simulates carbon cycle dynamics (both plant and soil) and accounts for the limitation of nutrient (i.e., nitrogen and phosphorus) availability for plant growth and the nutrient competition between plants and microbes (Burrows et al., 2020; Zhu et al., 2019; Golaz et al., 2019; Zhu et al., 2020). The ELMv1-ECA uses a Centurylike soil carbon decomposition cascade model with vertically resolved soil biogeochemistry (Koven et al., 2013), and explicitly accounts for the influence of substrate and nutrient availability on soil respiration (both root and microbes) (Zhu et al., 2019)." (Lines 703 - 707 in the revised manuscript)

"ELMv1-ECA considers the availability of carbon substrate as an important driver of methanogenesis activity and methane production (Riley et al., 2011; Xu et al., 2016)." (Lines 741 - 742 in the revised manuscript)

We would also like to conduct more analysis on soil carbon; however, due to the lack of site-level observations, we cannot compare simulated soil carbon with observation for this study. We will explore available *in situ* soil carbon datasets and incorporate these analyses in our future studies.

2) Snow. The main problem with ESM's underestimating winter soil temperature is often related to representation of snow, so I was surprised that this was not discussed in more detail, and only the phase change was considered as leading to underestimated winter soil temperatures (although snow is mentioned once in the results). For example, Burke et al (2020) show the offset between air and soil temperature in CMIP5 and CMIP6 ESM's as a function of snow depth. In models that poorly represent snow, the offset can be up to 10 degree C biased - meaning at 10 degree C cold bias in winter soil temperature. In models that improved their snow scheme between CMIP5 and CMIP6, there is a huge improvement in this. Even in a model that does not represent latent heat \*at all\*, the winter soil temperature offset against air temperature is substantially smaller than in models with a poor snow insulation scheme. I would therefore strongly recommend that snow is considered in terms of the simulation of winter soil temperatures. I suggest that at the point where snow depth is discussed in the results (see specific comments, below), an assessment of how well the snow is simulated should be presented with supplementary figures.

**R1C2:** We agree with the reviewer on the importance of accurately simulated snow conditions (e.g., snow thermal insulation and snow coverage-related impact). In our following paper (Tao et

al., 2021), we have conducted more experiments investigating how biases in simulated snow variables (i.e., snow depth, snow water equivalent (SWE), and snow coverage) are propagated to biases in soil temperature and soil moisture, and then translated into biases in heterotrophic respiration and cold-season  $CO_2$  and  $CH_4$  emissions over pan-Arctic permafrost regions. Please also see our response to Reviewer #2 (R2C1). For this study, the snow depth measurements at the study sites are problematic, showing about 30 cm snow depth during summer times (see raw data by Oechel and Kalhori (2018)). For instance, at ATQ, a site that shows the most reasonable snow depth observations, there are suspicious snow depth measurements in summer (Figure R1.1b). We also checked Snow Telemetry (SNOTEL) sites but did not find one close enough to our sites for a better comparison. Indeed, continuous quality snow measurements, especially SWE, are extremely challenging to obtain (Pirazzini et al., 2018; McGrath et al., 2019).

Still, as the reviewer suggested, we have provided comparison results of simulated and observed snow depth despite the suspicious measurements (Figure R1.1 as Figure S9 in the supplementary file); we also added discussion on how sensitive the simulated soil temperature and carbon fluxes are to snow depth in the supplementary file (Figure S9). We also added a sentence to the revised manuscript.

"Sensitivity analysis demonstrates large impacts of snow depth on simulated winter soil temperature, summer soil moisture, heterotrophic respiration, and CO2 fluxes (Figure S9)." (Lines 588 - 589 in the revised manuscript)

---

## Editor Decision (ED1)

Second review of "Improved ELMv1-ECA Simulations of Zero-Curtain Periods and Cold-season CH4 and CO2 Emissions at Alaskan Arctic Tundra Sites" Tao et al

With thanks to the authors for their detailed response and apologies for being late with my reply. Most of the major issues have been sorted but there are still some unclear parts, please see my comments below. **Note, all line numbers here are based on the marked up version.**

General comments

R1C1: "R1C1: As we pointed out in the manuscript (lines X - Y)," - line numbers missing here... Having dug into it I think that lines 127-128 in the marked up version suggest the carbon is estimated from the soil properties. But then it turns out in Section 3.2 that this is only for one site, so this should be mentioned when this sentence first appears ("In addition, we used ABoVE soil moisture measurements to derive site-scale soil porosity and organic carbon content *at IVO* (see Section 3.2) "). I would also add here, for clarity "which is used to prescribe thermal and hydraulic soil properties. Note that carbon substrate for respiration is simulated dynamically in the model - see Appendix B."

Section 3.2 states that a global soil C dataset is mostly used to derive the soil properties, but for IVO the porosity is used to estimate soil C, and it is also stated that "The derived SOC content is also consistent with the soil survey data reported in Davidson and Zona (2018)", which suggests to me that some soil carbon data is in fact available, at least for this site, contrary to what the authors have said in their response?

Lastly I would still like to see an acknowledgement somewhere (Summary/Discussion would be best) that the optimised decomposition functions would be biased if there is a bias in simulated soil carbon / substrate, and therefore should not be taken directly to other models without further analysis.

R1C2
Thanks for the response, I appreciate that the analysis of snow depth was added to the supplementary. However I still think it needs to be highlighted more carefully in the main text as an important controlling variable, to give a more complete picture for any reader who is not already an expert.

For example, in the introduction, on line 74 (marked up version!) you could add something along the lines of "We note that representation of snow can also play a major role in underestimation of winter soil temperatures [reference], although we do not focus on this process here."

In the discussion you added
"Sensitivity analysis demonstrates large impacts of snow depth on simulated winter soil temperature, summer soil moisture, heterotrophic respiration, and CO2 fluxes (Figure S9)." - I would definitely recommend adding something here, like "therefore the simulation of snow should be the subject of future investigations"

R1C3: Phase change efficiency
Line 785 (marked up version), start with something like "To improve this scheme, we can incorporate..." so it's clear that you're not still describing the existing model.

I appreciate that some more equations were added. Equations A7 and A8 are totally clear. Then I would expect to see something that looks like a differential of equation A3 appearing in the updated

version of $T^{(n+1)}$ (so, there should be a factor of $1/B$ somewhere...). I guess maybe you just didn't include the equation for calculating $T^{(n+1)}$. I think that would be helpful to add.

I have several queries around equation A11. The freezing point depression temperature does not appear anywhere in this equation, it still has $T_f$, and it has the phase change efficiency which does not relate either to this temperature or to the original equations (A7 and A8) that you are trying to solve. This phase change efficiency slows down the freezing/melting when it takes a smaller value, and for freezing a smaller value corresponds to less liquid water, which makes sense (although done properly, the freezing point depression should demand a large energy to freeze liquid water when there is not much left, so this would somehow be a double factor?). But for melting, a smaller value of phase change efficiency corresponds to a small amount of ice, which suggests that melting will slow down as it approaches small amounts of ice left in the soil, which to me does not make sense. When there are small amounts of ice left in the soil they will be all surrounded by unfrozen water and it will be easier to transfer energy into them. It would make more sense if the phase change efficiency was always proportional to the liquid water, and then it would somehow represent the freezing curve is a curve and takes more energy for freeze/thaw when there is less liquid water. But again, I am still not sure it is necessary if you properly follow the freeze curve.

How do you calculate $w\_ice^{(n+1)}$ in equation A11, is this going to be different from the previous model version because the latent heat was included in the original temperature change equation (A7/A8) ? This is a key thing, right?

I would request still more clarification of this phase change efficiency to make this paper clear.

R1C4: Thanks for these changes, all looks good!

R1C5 - justification for transport of methane through frozen soil / aerenchyma. In general this is clearer, thanks for the efforts on this. Just a couple more comments:

In the Appendix it describes epsilon_(snowdiff) (line 913), which was added but it does not show the equation to show how this parameter was applied. This would be helpful to show. Also the justification for the parameter choice, since I understand that this parameter was not varied in the sensitivity study.

Line 247: "We also conducted sensitive tests on seven CH4 parameterizations, including six parameterizations resulting from fractional three key variables and one parameterization scheme using all the tested values for the three variables "
In this sentence:
sensitive -> sensitivity
"fractional three key variables" does not make sense.
"tested values for the three variables" - I think you mean parameters, not variables? But even then, this part of the sentence is unclear.
Line 252: again 'sensitive' -> 'sensitivity'

R1C6 - adding IVO CO2 to analysis: thanks for doing this.
However in Section 4.2, the discussions of CO2 emissions between sites are mostly unchanged and do not include IVO, despite its being added to the plots, please check these and modify as necessary.
(For example, line 544-545 in marked up version: "Thus, the improved NSEs for CO2 and CH4 emissions at BES/CMDL and BEO were larger than those at ATQ" - and what about IVO?)

Line 650-656. IVO is still missing from this part also.

(New) Figure 5, the IVO plot is covered over by the legend. We should be able to see some values for CO2, at least? Looking at Figure 6 it looks like CO2 is significantly improved at IVO, so this should be apparent in the NSE for CO2?

R1C7 "we had checked the emissions vs. temperature and moisture (included in the authors' comments uploaded earlier)" Would this not be worth including in the manuscript / supplementary material?

"In the future, we will apply a Macromolecular Rate Theory (MMRT)-based temperature sensitivity approach, which uses a quadratic relationship to approximate the CH4 - temperature dependencies and thus can address the CH4 hysteresis effect (Chang et al. 2020, 2021) "
This implies simply changing the temperature function to a quadratic? Chang et al 2020 shows that the microbial dynamics are important for the seasonal hysteresis effect. Chadburn et al 2020 (https://agupubs.onlinelibrary.wiley.com/doi/full/10.1029/2020GB006678) also showed that the hysteresis effect can be captured by modelling methanogen seasonal dynamics, without MMRT. Therefore I am not sure that this is the key, but rather the fact that methanogens are slow-growing and slow responding organisms so they introduce a lag time on methane emissions. Thus, future work should consider simulating microbial population/activity levels.

Line 488-489 "this mechanism and wetland inundation dynamics together would cause hysteretic effects on CH4 emission response to soil temperatures". The use of 'this mechanism' implies that advective heat transport is the cause of hysteresis. If anything advective heat transport would cause thaw to happen more quickly in the early season. In fact the hysteresis is likely more related to the microbial activity level, or potentially the substrate distribution in the soil. Please clarify this.

Specific Comments

Introduction

Line 73 of marked up version: "CO2 emissions" -> "emissions of CO2" to link up with the "and CH4" that follows.

Data

"The CARVE CO2 measurements were not available at the data archive we used here"
Does this mean CARVE was only used for CH4? Then you should say "and >CH4 from< Carbon in Arctic Reservoirs Vulnerability Experiment (CARVE) flight campaign " in the previous sentence, that would make it a lot clearer.

"monthly winter-time CO2 flux data at the same towers assembled by Natali et al. (2019) are included to complement CO2 observations from 2013 to 2014"
This still does not make sense, if you already have CO2 observations from 2013 to 2014 which the Natali et al observation are complementing... where are they from? Do you mean "to *complete* the CO2 observations" ? Or "to complement CO2 observations from 2015 to 2017" ?

Line 123 "evaluating" -> "evaluation of"

Methods

Line 321 in marked up version: "Results vary with soil depths", does this mean "Results for ZCP duration vary with soil depth at which the ZCP is taken" ? Please replace if so, or clarify if not.

Results

R1C21 Response: "The pattern (i.e., lower soil moisture in shallow soil layers and higher soil water in deep layers) is shown in Figure 3 (Figure 2 in the revised manuscript) by the magenta vs. green lines during summertime when the active layers reach the deepest thaw depths."
If you look at Figure 2a (BES/CMDL), when the active layer is deepest the water is lower in every layer except the bottom one, but this bottom one I believe is partially frozen so it's not possible to tell how much water is actually in there? I guess it's just not totally clear without showing the unfrozen water as well, could you add the line for 'total water contents' as well as unfrozen, maybe in same colours but a different line style?

Line 519. "Figure 6 illustrates the uncertainty associated with the model representations of environmental influences on heterotrophic respiration and methane parameters"
Are you sure it's Figure 6? I think you might mean Figure S8, based on the discussion that follows.

Line 530 R1C25 "reasonably explained the varying influence along with the vertical soil profile (Niu and Yang, 2006)" This wording isn't clear, the varying influence of what on what? It would be great if you can rephrase this part.
Could you also say "(Niu and Yang 2006, Figure 1)" just to make that part clear, as you mentioned in the author response? Thanks! I am also struggling to see where there is a vertical soil profile in Niu and Yang Fig 1.

Summary

Line 704-705 "The underestimated emissions during post-ZCP months (Oct. to Nov.) are mainly caused by the lack of sudden bursts of CO2 and CH4 during the freeze-up period"
I don't think there was anything in the paper that showed this definitively (please correct me if I'm wrong). I suggest you tone this down to "may be caused by" instead of "are mainly caused by".

Appendix

Line 869 and 895: "on default" -> "by default"

R1C39 - thanks. I think there might still be an extra * in equation A10 (new version).

R1C42: Thanks for adding the reference. I have looked up "tiller" in Wania et al (2010) where they provide a footnote as to what it is, which indicates to me that perhaps it is not widely known. It might be helpful to include something similar to their footnote which I have copied here for convenience:
"Tillers are segmented stems produced at the base of many plants in the family Poaceae, with each stem possessing its own two-part leaf. The usage of the word "tiller" has been expanded to the order of Poales, which includes both groups, grasses (Poaceae) and sedges (Cyperaceae), and is here used in its wider meaning."

Figures

Thanks for the improvements to the Figures, they are definitely easier to interpret.

---

## Author Response (AR2)

We thank the reviewer for the constructive comments and suggestions. We have revised the manuscript further according to the reviewer's suggestions (see point-by-point responses below). Again, throughout this document, the reviewer's comments are reproduced in their entirety in black, and our responses are given directly afterward in blue. Line numbers here are based on the marked-up version of the revised manuscript.

Second review of "Improved ELMv1-ECA Simulations of Zero-Curtain Periods and Cold-season CH4 and CO2 Emissions at Alaskan Arctic Tundra Sites" Tao et al

With thanks to the authors for their detailed response and apologies for being late with my reply. Most of the major issues have been sorted but there are still some unclear parts, please see my comments below. **Note, all line numbers here are based on the marked up version.**

General comments

**R1C1:** "R1C1: As we pointed out in the manuscript (lines X - Y)," - line numbers missing here...Having dug into it I think that lines 127-128 in the marked up version suggest the carbon is estimated from the soil properties. But then it turns out in Section 3.2 that this is only for one site, so this should be mentioned when this sentence first appears ("In addition, we used ABoVE soil moisture measurements to derive site-scale soil porosity and organic carbon content *at IVO* (see Section 3.2) "). I would also add here, for clarity "which is used to prescribe thermal and hydraulic soil properties. Note that carbon substrate for respiration is simulated dynamically in the model - see Appendix B."

Thank you. As suggested, we have modified the sentence as below:

> "In addition, we used ABoVE observed maximum soil moisture to infer site-scale soil porosity and then organic carbon content at IVO (see Section 3.2), which is used to prescribe thermal and hydraulic soil properties. Note that carbon substrate for respiration is simulated dynamically in the model (see Appendix B)." (lines 123 - 125)

Section 3.2 states that a global soil C dataset is mostly used to derive the soil properties, but for IVO the porosity is used to estimate soil C, and it is also stated that "The derived SOC content is also consistent with the soil survey data reported in Davidson and Zona (2018)", which suggests to me that some soil carbon data is in fact available, at least for this site, contrary to what the authors have said in their response?

The soil survey data in Davidson and Zona (2018) are not quantitative SOC along with soil depth but are organic layer thickness. To clarify, we modified the sentence as below:

> "The derived SOC content is also consistent with the organic layer thickness reported in Davidson and Zona (2018)." (lines 262 - 263)

Lastly I would still like to see an acknowledgement somewhere (Summary/Discussion would be best) that the optimised decomposition functions would be biased if there is a bias in simulated soil carbon / substrate, and therefore should not be taken directly to other models without further analysis.

Thank you for the suggestion. We have added two sentences into the "5. Summary and Discussion" section:

"Note that the optimized parameterizations would be biased if there is a bias in simulated soil carbon, and therefore should not be taken directly to other models without further analysis. Instead, the optimization procedure described in this study provides a roadmap that can be directly adopted to calibrate other models at different sites." (lines 599 - 602)

**R1C2:** Thanks for the response, I appreciate that the analysis of snow depth was added to the supplementary. However I still think it needs to be highlighted more carefully in the main text as an important controlling variable, to give a more complete picture for any reader who is not already an expert.

For example, in the introduction, on line 74 (marked up version!) you could add something along the lines of "We note that representation of snow can also play a major role in underestimation of winter soil temperatures [reference], although we do not focus on this process here."

Thank you for the suggestion. We have added the sentence to the revised manuscript:

"We note that snow representation can also play a major role in correctly simulating winter soil temperatures (Slater et al., 2017; Lawrence and Slater, 2010), although we do not focus on this process here." (lines 70 - 72)

In the discussion you added "Sensitivity analysis demonstrates large impacts of snow depth on simulated winter soil temperature, summer soil moisture, heterotrophic respiration, and CO2 fluxes (Figure S9)." – I would definitely recommend adding something here, like "therefore the simulation of snow should be the subject of future investigations"

Thank you for the suggestion. We have modified the sentence as below:

"Sensitivity analysis demonstrates large impacts of snow depth on simulated winter soil temperature, summer soil moisture, heterotrophic respiration, and $CO_2$ fluxes (Figure S9); therefore, the simulation of snow should be the subject of future investigations." (lines 610 - 612)

**R1C3:** Phase change efficiency
Line 785 (marked up version), start with something like "To improve this scheme, we can incorporate..." so it's clear that you're not still describing the existing model.

Thank you for the suggestion. We have added this point to the revised manuscript:

> "To improve this scheme, we can incorporate soil-water freezing phase change into equation (Eq. A1) and rewrite the heat transfer equation as …" (lines 694 - 695)

I appreciate that some more equations were added. Equations A7 and A8 are totally clear. Then I would expect to see something that looks like a differential of equation A3 appearing in the updated version of $T^{n+1}$ (so, there should be a factor of 1/B somewhere...). I guess maybe you just didn't include the equation for calculating $T^{n+1}$. I think that would be helpful to add.

As Appendix A described, our model solves the heat transfer equation using the Crank-Nicholson method, which combines the explicit and the implicit method, and the numerical solution for $T_i^{n+1}$ is documented in detail in Oleson et al. (2013). The updated solution basically follows that framework, with modifications to the phase change treatment, as described in the Appendix. We trust it should be fine to only include equations that are impacted by our modifications, e.g., the updated $T_i^{n+1}$ (Eq. A11), instead of including many equations that are publicly available in the literature (Oleson et al., 2013).

I have several queries around equation A11. The freezing point depression temperature does not appear anywhere in this equation, it still has T_f, and it has the phase change efficiency which does not relate either to this temperature or to the original equations (A7 and A8) that you are trying to solve. This phase change efficiency slows down the freezing/melting when it takes a smaller value, and for freezing a smaller value corresponds to less liquid water, which makes sense (although done properly, the freezing point depression should demand a large energy to freeze liquid water when there is not much left, so this would somehow be a double factor?). But for melting, a smaller value of phase change efficiency corresponds to a small amount of ice, which suggests that melting will slow down as it approaches small amounts of ice left in the soil, which to me does not make sense. When there are small amounts of ice left in the soil they will be all surrounded by unfrozen water and it will be easier to transfer energy into them. It would make more sense if the phase change efficiency was always proportional to the liquid water, and then it would somehow represent the freezing curve is a curve and takes more energy for freeze/thaw when there is less liquid water. But again, I am still not sure it is necessary if you properly follow the freeze curve.

We thank the reviewer for these thoughts, and would like to make three related points. First, as Eq. A10 indicates, the freezing point depression temperature is expressed as a function of $T_f$, and therefore Eq. A11 does include $T_f$.

Second, we would like to clarify the distinction between soil water phase change and the associated latent heat. Since the soil water freezing process releases latent heat instead of demanding energy, it is not a double factor to employ the phase change efficiency.

Third, soil ice thawing requires energy. The phase change efficiency is applied to the initially estimated energy and mass change involved, i.e., $H_i$ and thus $H_m$. During the soil ice thawing process, the decreasing phase change efficiency as ice fraction decreases

means the process demands less and less energy for thawing further, as indicated by the reviewer. As a result, the new phase change scheme leads to better simulated soil temperatures than the baseline scheme during the thawing season, especially at IVO (red vs. blue in Figure 2). To indicate this improvement, we have added the following sentence to the revised manuscript:

> "Simulations with the new phase change scheme also show improved agreements between simulated and observed soil temperatures during the spring thawing season compared to the baseline results (red vs. blue in Figure 2)." (lines 407 - 408)

How do you calculate w_ice^(n+1) in equation A11, is this going to be different from the previous model version because the latent heat was included in the original temperature change equation (A7/A8) ? This is a key thing, right?

To better explain the updated mass of ice and liquid water, we have added the following sentence:

> " Here, $w_{ice,i}^{n+1}$ is calculated by (Eq. A5) as well, but with updated $H_m$ (i.e., $-\varepsilon_i c_i \frac{\Delta z_i}{L_f} ( Tv_i^{n+1} - T_i^{n+1} )$ )." (line 726)

I would request still more clarification of this phase change efficiency to make this paper clear.

Please see our responses above.

**R1C4:** Thanks for these changes, all looks good!
**R1C5** - justification for transport of methane through frozen soil / aerenchyma. In general this is clearer, thanks for the efforts on this. Just a couple more comments:
In the Appendix it describes epsilon_(snowdiff) (line 913), which was added but it does not show the equation to show how this parameter was applied. This would be helpful to show. Also the justification for the parameter choice, since I understand that this parameter was not varied in the sensitivity study.

Thank you for the suggestion. We have added the equation for snow resistance (Eq. C18) and related descriptions to the revised manuscript (lines 808 to 815).

Line 247: "We also conducted sensitive tests on seven CH4 parameterizations, including six parameterizations resulting from fractional three key variables and one parameterization scheme using all the tested values for the three variables "
In this sentence:
sensitive -> sensitivity
"fractional three key variables" does not make sense.
"tested values for the three variables" - I think you mean parameters, not variables? But even then, this part of the sentence is unclear.

Thanks. Sorry for the typo. We had meant "factorial" here. To better describe the sensitivity experiments, we have simplified this sentence in the revised manuscript, and clearly listed the tested parameterizations in Table S3 (see the updated Table S3):

"We also conducted sensitivity tests on three key parameters related to $CH_4$ oxidation and transport processes and tested seven parameterizations (Table S3)." (lines 227 - 230)

Line 252: again 'sensitive' -> 'sensitivity'

We have modified the word as suggested. Thanks.

R1C6 - adding IVO CO2 to analysis: thanks for doing this.
However in Section 4.2, the discussions of CO2 emissions between sites are mostly unchanged and do not include IVO, despite its being added to the plots, please check these and modify as necessary.
(For example, line 544-545 in marked up version: "Thus, the improved NSEs for CO2 and CH4 emissions at BES/CMDL and BEO were larger than those at ATQ" - and what about IVO?)

Thank you. We have added some discussion about $CO_2$ results at IVO:

"At IVO, although generally showing low NSEs for $CH_4$, some new simulations have improved $NSE_{CO_2}$ that are larger than 0.5 (Figure 5), compared with -0.3 for baseline. Indeed, the best result at IVO (with a $NSE_{CO_2}$=0.78) significantly improved the simulation of summer $CO_2$ sink compared to baseline result (Figure 6)." (lines 484 - 486)

Line 650-656. IVO is still missing from this part also.
(New) Figure 5, the IVO plot is covered over by the legend. We should be able to see some values for CO2, at least? Looking at Figure 6 it looks like CO2 is significantly improved at IVO, so this should be apparent in the NSE for CO2?

Thanks. We have moved the legend outside the plot for IVO to show the figure better. At IVO, the $CO_2$ results are significantly improved compared to baseline results, with some NSEs for $CO_2$ larger than 0.5. But the NSE for $CH_4$ is still low at this site. See the updated Figure 5.

R1C7 "we had checked the emissions vs. temperature and moisture (included in the authors' comments uploaded earlier)" Would this not be worth including in the manuscript / supplementary material?

Thanks. We now have included the figure in the supplementary as Figure S10.

[Figure]

Figure S1 - (A similar figure as Fig.3 in Zona et al. (2016)). Daily $CH_4$ emissions vs. soil temperatures at 12 cm at two sites. Similarly, as in Zona et al. (2016), we applied a 30-day averaging window to smooth the daily data to produce clear seasonal progressions. Shaded blue areas indicate zero-curtain periods, i.e., $[-0.75\ °C, 0.75\ °C]$. At BES/CMDL and IVO, observed seasonal progressions proceed in opposite directions (e.g., from black to green and then to red), while modeled seasonal progressions follow the same clockwise direction.

"In the future, we will apply a Macromolecular Rate Theory (MMRT)-based temperature sensitivity approach, which uses a quadratic relationship to approximate the CH4 - temperature dependencies and thus can address the CH4 hysteresis effect (Chang et al. 2020, 2021) "
This implies simply changing the temperature function to a quadratic? Chang et al 2020 shows that the microbial dynamics are important for the seasonal hysteresis effect. Chadburn et al 2020 (https://agupubs.onlinelibrary.wiley.com/doi/full/10.1029/2020GB006678) also showed that the hysteresis effect can be captured by modelling methanogen seasonal dynamics, without MMRT. Therefore I am not sure that this is the key, but rather the fact that methanogens are slow-growing and slow responding organisms so they introduce a lag time on methane emissions. Thus, future work should consider simulating microbial population/activity levels.

Thanks. We now have modified the sentence as below:

> "In the future, we will incorporate a representation of methanogen seasonal dynamics and simulate microbial population and activity levels to address the hysteresis of $CH_4$ emissions with temperature." (Lines 437 - 439)

Line 488-489 "this mechanism and wetland inundation dynamics together would cause hysteretic effects on CH4 emission response to soil temperatures". The use of 'this mechanism' implies that advective heat transport is the cause of hysteresis. If anything advective heat transport would cause thaw to happen more quickly in the early season. In fact the hysteresis is likely more related to the microbial activity level, or potentially the substrate distribution in the soil. Please clarify this.

Thanks. We have modified the sentences as below:

"Also, methanogen seasonal dynamics would cause hysteretic effects on $CH_4$ emission response to soil temperatures (Chang et al., 2020; 2021; Chadburn et al., 2020)." (lines 435 - 437)

Specific Comments

Introduction

Line 73 of marked up version: "CO2 emissions" -> "emissions of CO2" to link up with the "and CH4" that follows.

We have modified the sentence (Line 70) as suggested. Thanks.

Data

"The CARVE CO2 measurements were not available at the data archive we used here" Does this mean CARVE was only used for CH4? Then you should say "and >CH4 from< Carbon in Arctic Reservoirs Vulnerability Experiment (CARVE) flight campaign " in the previous sentence, that would make it a lot clearer.
"monthly winter-time CO2 flux data at the same towers assembled by Natali et al. (2019) are included to complement CO2 observations from 2013 to 2014"
This still does not make sense, if you already have CO2 observations from 2013 to 2014 which the Natali et al observation are complementing... where are they from? Do you mean "to *complete* the CO2 observations" ? Or "to complement CO2 observations from 2015 to 2017" ?

Thanks. We have modified the sentences as below to clarify the data availability better.

"We assembled daily observations of $CO_2$ and $CH_4$ fluxes from 2013 to 2017 at five eddy-covariance flux tower sites in Alaska's North Slope tundra (Figure 1) from the Arctic-Boreal Vulnerability Experiment (ABoVE) project (2015 - 2017) (Oechel and Kalhori, 2018) and $CH_4$ fluxes from the Carbon in Arctic Reservoirs Vulnerability Experiment (CARVE) flight campaign (2013 - 2014) (Zona et al., 2016). The CARVE $CO_2$ measurements were not available at the data archive we used here; therefore, monthly winter-time $CO_2$ flux data from 2013 to 2014 at the same towers assembled by Natali et al. (2019) are included to complete $CO_2$ observations." (Lines 92 - 97)

Line 123 "evaluating" -> "evaluation of"

We have modified this (line 120) as suggested. Thanks.

Methods

Line 321 in marked up version: "Results vary with soil depths", does this mean "Results for ZCP duration vary with soil depth at which the ZCP is taken" ? Please replace if so, or clarify if not.

Thanks. We have modified the sentence (line 295) as suggested.

Results

R1C21 Response: "The pattern (i.e., lower soil moisture in shallow soil layers and higher soil water in deep layers) is shown in Figure 3 (Figure 2 in the revised manuscript) by the magenta vs. green lines during summertime when the active layers reach the deepest thaw depths." If you look at Figure 2a (BES/CMDL), when the active layer is deepest the water is lower in every layer except the bottom one, but this bottom one I believe is partially frozen so it's not possible to tell how much water is actually in there? I guess it's just not totally clear without showing the unfrozen water as well, could you add the line for 'total water contents' as well as unfrozen, maybe in same colours but a different line style?

Indeed, the moisture saturation Sf ($\theta_{liq}/\theta_{sat}$) shown in Figure 2 means unfrozen (liquid) water content. We have stated this in the revised manuscript, and now we also clarified this in the figure caption by adding, "Here, the moisture saturation means soil unfrozen (liquid) water content." (Line 1134)

Line 519. "Figure 6 illustrates the uncertainty associated with the model representations of environmental influences on heterotrophic respiration and methane parameters"
Are you sure it's Figure 6? I think you might mean Figure S8, based on the discussion that follows.

Thanks. We have changed Figure 6 to Figure S8.

Line 530 R1C25 "reasonably explained the varying influence along with the vertical soil profile (Niu and Yang, 2006)" This wording isn't clear, the varying influence of what on what? It would be great if you can rephrase this part.
Could you also say "(Niu and Yang 2006, Figure 1)" just to make that part clear, as you mentioned in the author response? Thanks! I am also struggling to see where there is a vertical soil profile in Niu and Yang Fig 1.

Thanks. Fig. 1 in Niu and Yang (2006) shows that the relationship between unfrozen soil moisture and soil temperature varies with clay fraction in soils, which reflects the vertical distribution of soil properties along with soil depth. We have modified this sentence as below:

"…, reasonably explained the varying influence along with the vertical soil profile (i.e., relationships between soil liquid water content and soil temperature varies with soil clay fraction as demonstrated by Fig.1 in Niu and Yang, 2006)." (Lines 468 to 469)

Summary

Line 704-705 "The underestimated emissions during post-ZCP months (Oct. to Nov.) are mainly caused by the lack of sudden bursts of $CO_2$ and $CH_4$ during the freeze-up period" I don't think there was anything in the paper that showed this definitively (please correct me if I'm wrong). I suggest you tone this down to "may be caused by" instead of "are mainly caused by".

Thanks. We have changed "are mainly caused by" to "may be caused by" (Line 614).

Appendix

Line 869 and 895: "on default" -> "by default"

Modified as suggested. Thanks.

R1C39 - thanks. I think there might still be an extra * in equation A10 (new version).

Removed the extra *. Thanks.

R1C42: Thanks for adding the reference. I have looked up "tiller" in Wania et al (2010) where they provide a footnote as to what it is, which indicates to me that perhaps it is not widely known. It might be helpful to include something similar to their footnote which I have copied here for convenience:
"Tillers are segmented stems produced at the base of many plants in the family Poaceae, with each stem possessing its own two-part leaf. The usage of the word "tiller" has been expanded to the order of Poales, which includes both groups, grasses (Poaceae) and sedges (Cyperaceae), and is here used in its wider meaning."

Thank you very much. We now have included a sentence here to better explain "tiller":

> "Here, tillers mean segmented stems of plants in the Order of Poales, including grasses (Poaceae) and sedges (Cyperaceae) (Wania et al., 2010)." (Lines 790 to 791)

Figures
Thanks for the improvements to the Figures, they are definitely easier to interpret.

Thanks.

**Reference**

Chadburn, S. E., Aalto, T., Aurela, M., Baldocchi, D., Biasi, C., Boike, J., Burke, E. J., Comyn-Platt, E., Dolman, A. J., Duran-Rojas, C., Fan, Y. C., Friborg, T., Gao, Y., Gedney, N., Gockede, M., Hayman, G. D., Holl, D., Hugelius, G., Kutzbach, L., Lee,

H., Lohila, A., Parmentier, F. J. W., Sachs, T., Shurpali, N. J., and Westermann, S.: Modeled Microbial Dynamics Explain the Apparent Temperature Sensitivity of Wetland Methane Emissions, Global Biogeochemical Cycles, 34, 2020.

Chang, K. Y., Riley, W. J., Crill, P. M., Grant, R. F., and Saleska, S. R.: Hysteretic temperature sensitivity of wetland CH4 fluxes explained by substrate availability and microbial activity, Biogeosciences, 17, 5849-5860, 2020.

Chang, K. Y., Riley, W. J., Knox, S. H., Jackson, R. B., and al., e.: Substantial hysteresis in emergent temperature sensitivity of global wetland CH4 emissions, (Under Review), 2021.

Davidson, S. J., and Zona, D.: Arctic Vegetation Plots in Flux Tower Footprints, North Slope, Alaska, 2014, ORNL DAAC, Oak Ridge, Tennessee, USA. https://doi.org/10.3334/ORNLDAAC/1546, 2018.

Lawrence, D. M., and Slater, A. G.: The contribution of snow condition trends to future ground climate, Clim Dynam, 34, 969-981, 2010.

Oleson, K. W., Lawrence, D., Bonan, G., Drewniak, B., Huang, M., Koven, C., Levis, S., Li, F., Riley, W., and Subin, Z.: Technical Description of version 4.5 of the Community Land Model (CLM)(NCAR Technical Note No. NCAR/TN-503+ STR). Citeseer, National Center for Atmospheric Research, PO Box, 3000, 2013.

Slater, A. G., Lawrence, D. M., and Koven, C. D.: Process-level model evaluation: a snow and heat transfer metric, Cryosphere, 11, 989-996, 2017.

Wania, R., Ross, I., and Prentice, I. C.: Implementation and evaluation of a new methane model within a dynamic global vegetation model: LPJ-WHyMe v1.3.1, Geosci Model Dev, 3, 565-584, 2010.